# Descriptions and Barcoding of Five New Chinese *Deuterophlebia* Species Revealing This Genus in Both Holarctic and Oriental Realms (Diptera: Deuterophlebiidae) [note 1]

**DOI:** 10.3390/insects13070593

**Published:** 2022-06-28

**Authors:** Xuhongyi Zheng, Zhiteng Chen, Pengxu Mu, Zhenxing Ma, Changfa Zhou

**Affiliations:** 1The Key Laboratory of Jiangsu Biodiversity and Biotechnology, College of Life Sciences, Nanjing Normal University, Nanjing 210023, China; zxhy2000@outlook.com (X.Z.); euplectmpx@163.com (P.M.); a952718141@outlook.com (Z.M.); 2School of Grain Science and Technology, Jiangsu University of Science and Technology, Zhenjiang 212004, China; wstcczt@outlook.com

**Keywords:** biodiversity, China, *Deuterophlebia*, mountain midge, new species, taxonomy

## Abstract

**Simple Summary:**

The insects of the dipteran family Deuterophlebiidae have remarkedly specialized morphology and biology. Larvae of them prefer to live in the rapid current of monte streams. Their adults, with extremely elongated antennae and enlarged thorax covering their head, were recorded having few hours lifespan. The research on this group of peculiar insects began one century ago and 14 species have been reported from Asian Palearctic and Nearctic realms. However, the species of vast China have not been collected or studied systematically so far, only one species was known while the molecular work did not appear. Here, with specific sampling widely in China and careful classification, we provide the first species diversity data of this country, the first interspecific genetic distance, and the first Oriental record of the family. As a result, five new species are added to the previous eight Asian known ones, distribution and biology of this genus are updated correspondingly.

**Abstract:**

The monotypic family Deuterophlebiidae of China was recorded twice previously from far northwest upon adults, the most parts of this country have not been investigated, leaving a huge blank of knowledge on their morphology, diversity, biology, or distribution. After deliberated collecting and rearing in recent years, we obtained more than one thousand specimens of Deuterophlebiidae, they are classified into five new species herein: *Deuterophlebia sinensis* sp. nov., *D. yunnanensis* sp. nov., *D. wuyiensis* sp. nov., *D. acutirhina* sp. nov. and *D. alata* sp. nov. Detailed descriptions and photographs of gathered life stages are given for these new species. Adults of them can be identified by chaetotaxy and length ratio of flagellomeres and legs, microtrichia on postgena and shape of their clypeus, pupae can be recognized by thoracic spines and abdominal chitin bands, and larvae can be separated by setae on thorax and abdomen. Genetic distances between species are 0.086–0.175 based on their COI genes. This contribution represents the first database of the enigmatic Deuterophlebiidae from China and shows a new distribution pattern of *Deuterophlebia*. In addition, the discovery throws some light on the origin and biogeography of the genus and family.

## 1. Introduction

The torrential dipteran family Deuterophlebiidae (mountain midges) comprises the single Holarctic genus *Deuterophlebia* Edwards, 1922 with 14 described species [1,2,3,4,5]. These mountain midges are found rarely in most regions because of their specialized habits, tiny and fragile bodies, and short-lived adults, species were recorded scattered across Asian Russia, Kyrgyzstan, India, Afghanistan, Mongolia, Korea, and Japan [3,4,5,6,7], and North America (from Canada and the United States) [2,8,9,10,11,12]. In China, species of Deuterophlebiidae were only reported twice by two researchers [3,13], and no other reports are available from the past three decades. The outline of Chinese deuterophlebiid fauna has not been presented until now.

The Palaearctic *Deuterophlebia* species are distributed from the Himalayas in the west to Japan in the east, from northern parts of Russia in the north to Sikkim in the south [3] but remain less known in China. The vast central geographical gap between the Palaearctic species and the limited knowledge from China hinders further biogeographic research.

It was hypothesized that the ancestor of Deuterophlebiidae originated from the Indian Himalayas first (here have two plesiomorphic species *D. brachyrhina* Courtney, 1994 and *D. oporina* Courtney, 1994), then dispersed twice to the Nearctic region, including the invasion of a single species, *D. inyoensis* Kennedy, 1960 and a subsequent invasion of the common ancestor of the other five extant species [3]. If more knowledge of Chinese Deuterophlebiidae is obtained, we may recover more accurate historical migration routes and know more speciation process of this interesting insect group.

Since the establishment of the family with two male adults in 1922 [14], most adults of the eight Asian *Deuterophlebia* species have been described except for *Deuterophlebia tyosenensis* Kitakami, 1938 known only by immatures [3,4], but starting from the work of Pulikovsky in 1924 [15], pupae and larvae of only four species have been reported [3,4]. Studies on microstructures of different stages proved that the larval and pupal characteristics are equally important as adult characteristics and sometimes even more informative [16,17,18]. Morphology of more life stages of more species will be helpful to understanding the history and character or behavioral evolution of this family. 

Deuterophlebiidae was believed to have a unique position in the phylogenetic tree of Diptera and was considered to be one of the most basal branches in some research based on both morphological and molecular characters [19,20,21,22,23,24]. Some new methods and data, such as the mitochondrial genome, are also used in recent years [25]. Nevertheless, only one deuterophlebiid species has a sequence listed in the GenBank, and the genetic diversity in the family has never been mentioned before. The fresh Chinese materials of Deuterophlebiidae will provide not only new character states but also new opportunities to obtain more genetic data on this issue.

In the past few years, we collected more than one thousand larvae, pupae, and adults of *Deuterophlebia* from Sichuan, Yunnan, and Fujian provinces, China. Morphologically, five new species are recognized and all COI of them are also sequenced to test their genetic distances molecularly. They show China hosts more diverse deuterophlebiids and presents a new distribution map of this family.

## 2. Materials and Methods

Larvae and pupae were collected from the surface of stones underwater in the section of rapids. Adults were obtained from three sources: bodies trapped by spider webs, bodies floating on banks of creeks, or living adults reared by cages set in the stream. Specimens were examined under a stereomicroscope (Nikon SMZ 745T, Nikon, Tokyo, Japan). Details of small structures were studied by dissection and treatment in 10% NaOH (30 °C, 30 min), and observed and photographed in ethanol with a camera (Nikon 50i, Nikon, Tokyo, Japan) mounted on a microscope. The heads of adults were also photographed by a Scanning Electron Microscope (Apreo 2S, Thermo Fisher Scientific Company, Waltham, MA, USA). Terminology mainly follows that of Courtney (1994) [3]. One male adult paratype of *D. sinensis* sp. nov., *D. yunnanensis* sp. nov., *D. acutirhina* sp. nov., and one female adult paratype of *D. wuyiensis* sp. nov. mounted into slides by neutral balsam, deposited in the Diptera collection of College of Life Sciences, Nanjing Normal University. Other specimens were preserved in 85% ethanol and deposited in the Diptera collection of the College of Life Sciences, Nanjing Normal University and School of Grain Science and Technology, Jiangsu University of Science and Technology. 

Due to the lack of previous molecular research in this family, we refer to the previous experimental conditions and means of our laboratory [26,27]. Total genomic DNA was extracted from the abdomen of specimens (head, legs, and terminalia not destructed) using Animal Genomic DNA Kit (TsingKe Biotech Co., Beijing, China). The mitochondrial genes cytochrome c oxidase subunit I was PCR-amplified using the Premix Taq (Takara Bio Inc., Beijing, China) with the forward primer F (50-TTC AGC CAC TTT ACC GCG-30, [28]) and reverse primer HCO2198 (50-TAA ACT TCA GGG TGA CCA AAA AAT CA-30, [29]). PCR conditions included initial denaturation at 94 °C for 5 min, 40 cycles of denaturation at 94 °C for 30 s, annealing at 50 °C for 30 s, and extension at 72 °C for 40 s, with a final extension at 72 °C for 10 min. All sequences with GenBank accession number and specimen information are listed in Table 1. They were aligned using Muscle, and the K2P genetic distance was estimated in MEGA7 [26,27].

## 3. Results

Morphology of five new species *Deuterophlebia sinensis* sp. nov., *D. yunnanensis* sp. nov., *D. alata* sp. nov., *D. acutirhina* sp. nov. and *D. wuyiensis* sp. nov. are listed sequentially in Figure 1, Figure 2, Figure 3, Figure 4, Figure 5, Figure 6, Figure 7, Figure 8, Figure 9, Figure 10, Figure 11, Figure 12, Figure 13, Figure 14, Figure 15, Figure 16, Figure 17, Figure 18, Figure 19, Figure 20 and Figure 21, Figure 22 and Figure 23 are comparisons of adult heads and pupae of different species respectively.

### 3.1. Deuterophlebia sinensis sp. nov.

Description of male adults (Figure 1, Figure 2, Figure 3 and Figure 22A): Body length 2.2–2.5 mm (measurements here and below based on 5 specimens), uniformly dark brown (Figure 1A). Head dark brown, extremely flat, width 0.43–0.48 mm, obscured by prothorax in dorsal view (Figure 1A–C and Figure 22A). The median clypeal lobe convex, bearing around 20 setae (Figure 1B,C and Figure 22A). Compound eyes glabrous, the distance between eyes about 2.0× maximum diameter of one eye (Figure 1B). Microtrichia covered dorsal head (Figure 1B). Vestigial mouthparts in the form of an invaginated tubule, postgena with microtrichia, oral region glabrous. Ventral posterior edge of oral region (or mouth opening) sclerotized and ridged, convex near midline forming a mental tooth. A pair of tentorial pits present on each side of the oral region (Figure 1C and Figure 22A).

Antenna 9.0–10.3 mm, including scape, pedicel, and four flagellomeres (Figure 1B). Scape with microtrichia on the surface, slightly broader than pedicel and about 2.0× length of the pedicel or 1/3× head width. Pedicel globular, about 0.08 mm in length. First flagellomere sub-equal to basal two segments in length, with a subapical tubercle bearing 4–7 digitiform setae. Flagellomeres II–III similar in shape and length, half the length of flagellomere I, with similar subapical tubercles respectively to flagellomere I (Figure 1B,D). Flagellomere IV about 18× combined length of five basal antennal articles, and about 4× body length or 120× length of pedicel; narrowed gradually, with fine hair-like setae on the anterior side of basal half, setae on apical half clustered, with four or five clusters in total, also some setae at the slightly expanded apex (Figure 1D–F).

Thorax dark brown, covered with microtrichia. Pronotum in form of two small lobes at anterolateral mesothoracic angles. Mesonotum strongly expanded forward and downward (Figure 1A). Wings 4.8–5.4 mm in length, gray and semi-transparent; cubital area greatly enlarged. Anal margin and posterior half outer margins fringed with soft hair-like setae. The veins pale and inconspicuous, and the costal margin slightly thickened (Figure 3A).

Legs similar in shape, dark brown, chaetotaxy similar, with five types of setae: (1) microtrichia, densely covered on coxae, trochanters, and dorsal side of other segments; (2) bent sharp macrotrichia, sparsely on the dorsal margin of femora and proximal tibiae; (3) long soft capitate setae, on distal half of ventral edge of all tibiae, around the top of fore- and midtibiae, and ventral side of all tarsi; (4) long straight capitate setae, only present on empodium, arranged radially; (5) short digitiform setae, 1–3 pairs on each tarsomere, mingled in the capitate setae (Figure 2).

**Figure 1 insects-13-00593-f001:**
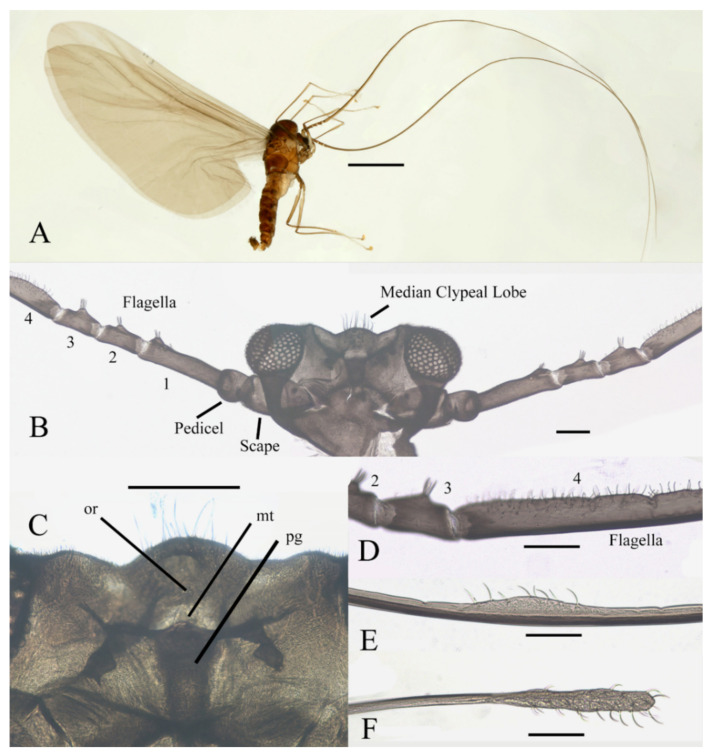
Male adult of *D. sinensis* sp. nov.: (**A**) male habitus (lateral view); (**B**) head (dorsal view); (**C**) oral region (ventral view); (**D**) flagellomeres; (**E**) middle part of terminal flagellomere; (**F**) apex of flagellomere IV. Abbreviation: or, oral region; mt, mental tooth; pg, postgena. Scale bars: (**A**) = 1.0 mm; (**B**–**F**) = 0.1 mm.

**Figure 2 insects-13-00593-f002:**
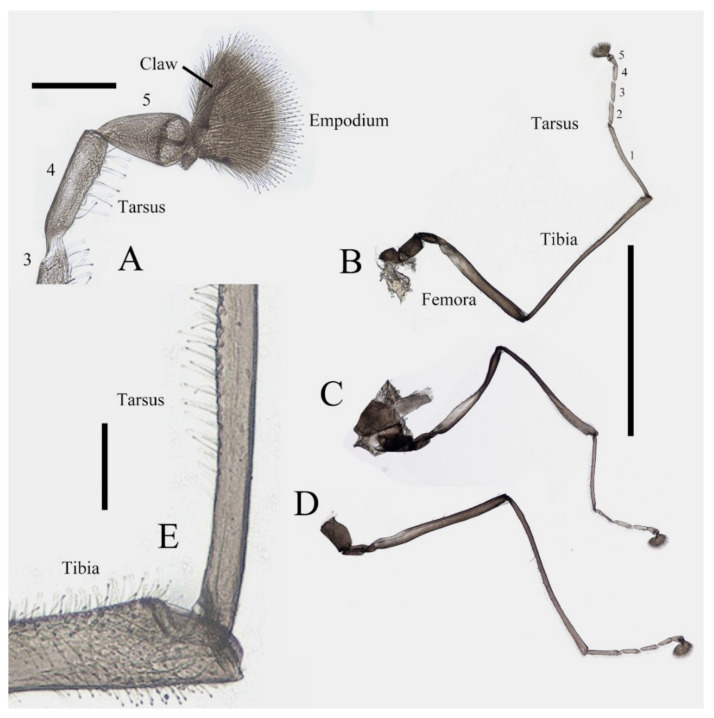
Legs of male *D. sinensis* sp. nov.: (**A**) claw of foreleg (ventral view); (**B**) foreleg; (**C**) midleg; (**D**) hindleg; (**E**) joint of the tibia and tarsus of foreleg. Scale bars: (**A**,**E**) = 0.1 mm; (**B**–**D**) = 1 mm.

Coxae of all legs much broader than trochanters, about 2.0× latter in length. Femora:tibiae:tarsi of forelegs = 4.0:6.0:5.0; forefemora flattened, tarsomere I:II:III:IV:V = 4.0:1.0:1.0:1.0:0.7, segment I–IV cylindrical; segment V conical and glabrous; empodium shell-shaped; claw sharp, covered by empodium (Figure 2B). Midleg shortest among all legs, length ratio and other features similar to foreleg (Figure 2C). Femora:tibiae:tarsi of hindleg = 5.4:6.0:3.4; tarsomere I:II:III:IV:V = 2.0:1.0:1.0:1.0:0.7 (Figure 2D).

Abdomen light to dark brown with nine segments, tapering posteriorly. Abdominal surface covered with microtrichia, visible tergites on segment III–VII; posterolateral angles of tergites blunt. The first two abdominal segments paler and shorter than the others. Segment VIII reduced to a chitin ring (Figure 1A and Figure 3B,D).

Sternite IX almost glabrous, fused with the dorsal plate. The dorsal plate parallel-sided, with a V-shaped apical emargination in varying degrees among individuals, with several sharp curved bristles. Gonostylus longer than the dorsal plate, broad at the base, rounded apically, flexor surface with numerous curved sharp bristles. Aedeagus in the form of a smooth tube, a length of about 3× width and equal to the gonostylus, extending beyond the dorsal plate (Figure 3C,D).

Description of female adults (Figure 4 and Figure 22B): Body length 2.2–2.4 mm (measurements here and below based on 2 female adults and 2 mature females dissected from pupae), besides sexual differences, generally similar to the male except following features of head and legs (Figure 4A).

The oral region located near the anterior margin of the head, postgena with microtrichia, and the oral region glabrous. Compound eyes more prominent than the male’s (Figure 4D). The antenna about 0.5 mm in length. The scape and pedicel similar to the male, but with several sharp setae on the pedicel. The length ratio of the antennal article I:II:III:IV:V:VI = 2.0:1.0:3.0:1.0:1.3:1.0, four flagellomeres narrowed basally and expanded apically. Flagellomeres I–III bearing 2–5 digitiform setae on or around the slight anteroventral prominences. Flagellomere IV with 2–4 sharp setae (Figure 4C,D).

**Figure 3 insects-13-00593-f003:**
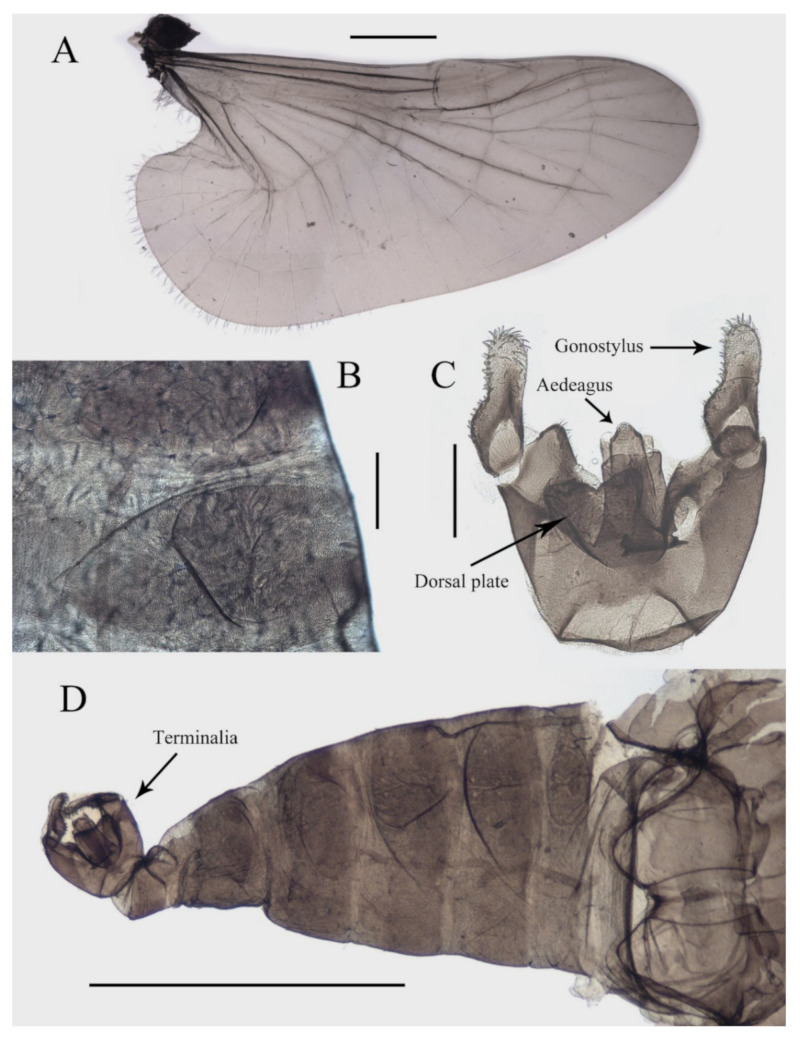
Structures of male *D. sinensis* sp. nov.: (**A**) wing; (**B**) abdominal tergites (lateral view); (**C**) terminalia (dorsal view); (**D**) abdomen (dorsal view). Scale bars: (**A**,**D**) = 1.0 mm; (**B**,**C**) = 0.1 mm.

Legs similar; femora: tibiae: tarsi of legs = 1.0:2.0:1.0. Femora wide and flat; tibiae cylindrical; length ratio of the foreleg and midleg tarsomere I:II:III:IV:V = 3.0:1.0:1.0:1.0:3.0; tarsomere I of hindlegs shorter than half of segment V; claws paired, stout and curved, with a sharp protrusion in the middle; empodium forming a long and hairy spine, more than half the length of the claw (Figure 4A,D). Legs with similar chaetotaxy, exhibiting three types of setae: (1) microtrichia, on the dorsal margin of tibiae; (2) bent sharp macrotrichia, sporadically set on femora and tibiae; (3) digitiform setae, sparsely set on the ventral side of all tarsi, 1–3 pairs on each segment (Figure 4A,D).

**Figure 4 insects-13-00593-f004:**
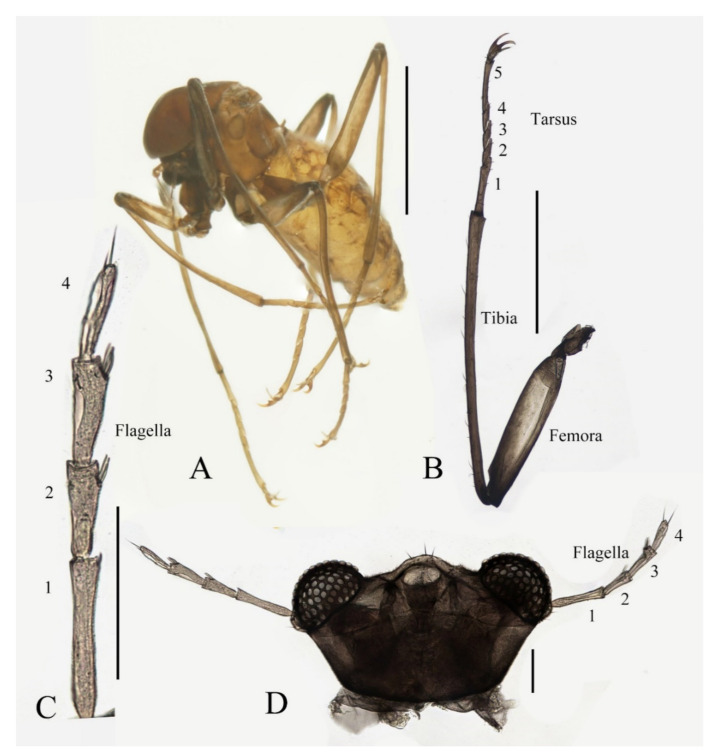
Female adult of adult *D. sinensis* sp. nov.: (**A**) female habitus (lateral view); (**B**) foreleg; (**C**) flagellomeres; (**D**) head (ventral view). Scale bars: (**A**,**B**) = 1.0; (**C**,**D**) = 0.1 mm.

Description of male pupae (Figure 5 and Figure 23A,B): Total length 2.5–2.8 mm (measurements here and below based on 4 specimens), width 1.7–1.9 mm, flattened, dorsal integument light to dark brown, densely covered with brown dots, divided into 11 distinct segments (Figure 23A,B).

Prothorax fused with mesothorax, forming a conical segment. A median suture and two pairs of dark dots present on the mesothorax. Mesothoracic margins each with a pair of spines and a gill (Figure 23A,B). Spines slightly curved, dark brown, length about half the width of metathorax, originated from a round base (Figure 5B). Ventral gills dark brown, each including four filaments: posterior filament single and short, pointing backward; anterior three gill filaments on a common base, similar in shape, slender and twisted. The second and third filaments fused at the base (Figure 5A). The metathorax narrow and completely surrounded by the mesothorax and first abdominal segment. Both meso- and meta-thoracic segments with paired transverse chitin bands, mesothoracic bands about 3× the length of those on metathorax (Figure 23A,B).

**Figure 5 insects-13-00593-f005:**
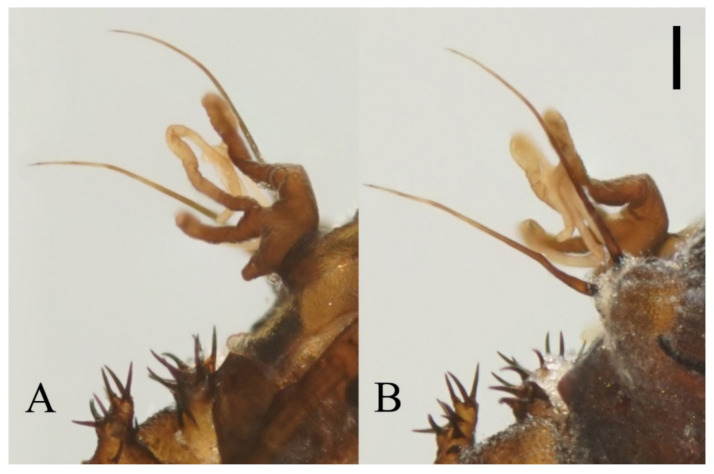
Male pupa of *D. sinensis* sp. nov.: (**A**) gill (ventral view); (**B**) thoracic spines (dorsal view). Scale bar = 0.1 mm.

Abdominal segments I and II similar, each with a pair of anterolateral projections, pointing down and forward respectively; each projection bearing 6–14 spines. Lateral margins of segments VI and VII with several spines on both dorsal and ventral sides, number and pattern of spines variable. Segment VIII shield-shaped, surrounded by segments VII and IX. Posterolateral projections of segments VII and IX directed medially (Figure 23A,B).

Adult structures visible on the ventral side. The head present directly below the mesothorax; antennal sheaths extended from both sides and surrounding body 2.0 times, forming a large elliptic ring lying on the ventral surface. Six leg sacs extended to the posterior of the antennal ring and expanded apically. Three pairs of black adhesive discs present on the ventral edges of abdominal segments III–V (Figure 23B).

Description of female pupae (Figure 23C,D): Total length 2.1–2.4 mm (measurements here and below based on 4 specimens), width 1.5–1.8 mm. Dorsal morphology similar to males except for smaller body size and mesothorax. Gender can be easily identified by the length of antennal sheaths, much thinner and shorter than males, only extended to wing sheaths. The apex of female leg sheaths only slightly expanded (Figure 23C,D).

Description of mature larvae (Figure 6, Figure 7 and Figure 8): Total length 4.0–6.0 mm (measurements here and below based on 4 specimens). The dorsal color of the body gray to light brown, and the abdomen translucent, divided into eleven segments by deep incisions (Figure 6). Head free and distinct, width 0.50–0.62 mm, mainly dark brown. The clypeus uniformly dark brown. The posterior edge sharply protruded backward for about half the length of the head capsule. The head surface almost glabrous, except for a patch of microtrichia near the antennal base. Eyes on both sides of the head (Figure 7A). Antenna 1.3–1.5 mm, two-segmented, originated from the anterolateral side of the head, with a black ring at base; proximal segment dark brown, glabrous, length about 3× diameter; distal article bifurcated, pale, scattered with dark dots; ventral branch slightly shorter than scape, with a blunt apex; dorsal branch three to four times the length of the proximal segment, tapered and sharp distally (Figure 6 and Figure 7A). Clypeus dark brown with an acute apex pointing forward, covering mouthparts, about half the length of head shell. Mouthparts form a tubular structure. Labrum tube-like, opening forward, with a row of long bristles on both sides and several spines on the central area. Mandibles and maxillae mostly hidden in the head shell. Mandibles divided into a curved structure and with a tuft of long bristles, each edge with neat bristles forming a comb-like structure. Maxillae have a setaceous circle around the front surface; palpi simplified and jointless (Figure 7B).

**Figure 6 insects-13-00593-f006:**
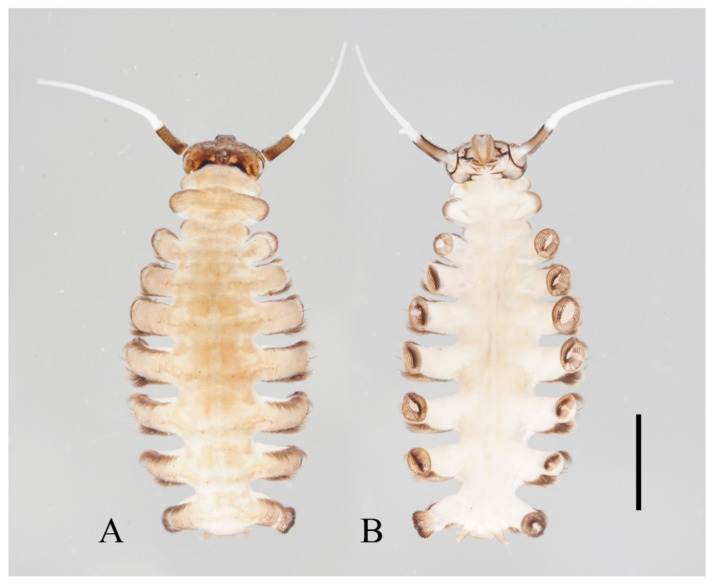
Larva of *D. sinensis* sp. nov.: (**A**) dorsal view; (**B**) ventral view. Scale bar = 1.0 mm.

First, three segments behind the head without prolegs, respectively regarded as pro-, meso- and meta-thorax. Mesothorax widest, metathorax narrowest. When pupa developed inside the body, gills, and spines can be identified. Abdomen eight-segmented, segments I–VII each with a pair of lateral prolegs (Figure 6). Anterior and posterior margin of each leg with a row of dense hairbrushes (Figure 7D). The dorsal surface of the thorax and abdomen scattered with various setae (sensilla), roughly divided into 4 types: (1) digitiform setae, simple or branched, on the thorax and abdominal segments; (2) long spine-like setae, only one near the top of each pseudopodium and tip of caudal appendages; (3) hair-like setae, simple or branched, on the thorax and abdominal segments; (4) short sharp conical setae, branched, only on caudal appendages (Figure 7C,D and Figure 8). Chaetotaxy of the unilateral body as in Figure 8: prothorax: with two bifurcated digitiform setae, hair-like setae on midline side; mesothorax similar to prothorax; I–IV abdominal segment: with three simple digitiform setae, between the outer two were two simple hair-like setae and a bifurcated or trifurcated hair-like setae between them; last abdominal segment and caudal appendage: four short sharp conical setae, terminal one hexafurcated, others trifurcated.

**Figure 7 insects-13-00593-f007:**
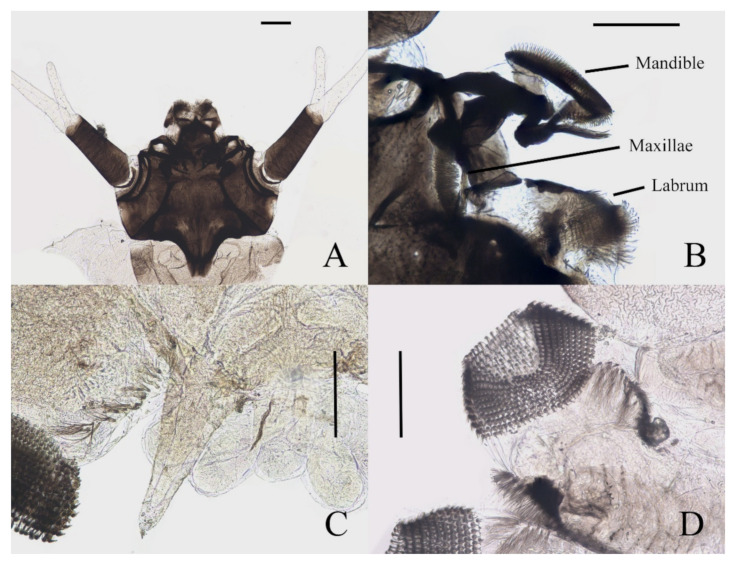
Larval structures of *D. sinensis* sp. nov.: (**A**) head enlarged (dorsal view); (**B**) mouthparts (dorsal view); (**C**) terminal of larva abdomen (ventral view); (**D**) prolegs (ventral view). Scale bars = 0.1 mm.

Prolegs apically sucker-like, with 9–13 rows of tiny claws. The claw with a ring of crochet at the apex (Figure 7D). Dorsal surface of larvae with sparse bristles. Last segment of the abdomen with a pair of caudal appendages; caudal appendages apically with a long single spine and two clusters of three digitiform setae around the spine. Five translucent anal papillae present on the ventral surface of the last segment (Figure 6 and Figure 7C,D).

**Figure 8 insects-13-00593-f008:**
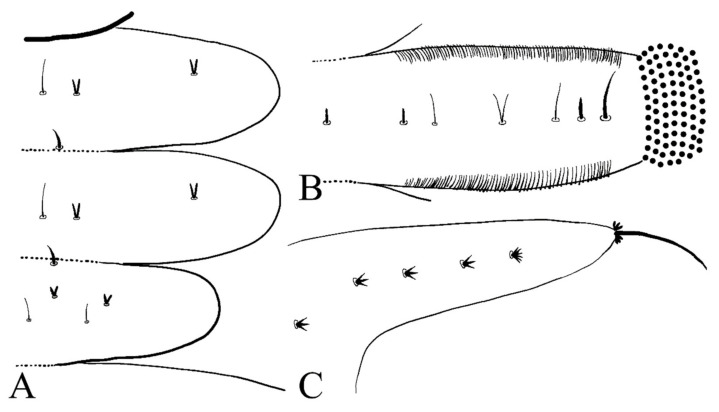
Larval chaetotaxy of *D. sinensis* sp. nov.: (**A**) thorax (dorsal view); (**B**) proleg (dorsal view); (**C**) caudal appendage (dorsal view).

Material Examined: Holotype: male adult, China: Sichuan Province, Kangding City, Mugecuo, 30°11′36.33″ N, 101°54′38.59″ E, 2860 m a.s.l., 24–28.VII.2021, Peng-Xu Mu, Xu-Hong-Yi Zheng, Zhi-Teng Chen leg. Paratypes: 90 males, 2 females, 6 male pupae, 6 female pupae, 10 larvae, same locality and data as holotype. Other material examined: 10 males, 3 pupae, 2 larvae, China: Sichuan Province, Li County, Miyaluo scenic spot, 31°41′21.34″ N, 102°44′35.97″ E, 2943 m a.s.l., 1–3.VIII.2021, Peng-Xu Mu, Xu-Hong-Yi Zheng, Zhi-Teng Chen leg.

Diagnosis: The new species *D. sinensis* sp. nov. is consistent with other Palaearctic species in its tri-furcated apical sensilla on the larval abdomen (Figure 8C). However, larvae of *D. sinensis* sp. nov. have a uniformly dark head, bear simple subtergal sensilla, and prelaterotergal sensilla on mesothorax (Figure 6 and Figure 8). Upon those, they can be easily distinguished from *D. sajanica* Jedlicka & Halgos, 1981, *D. nipponica* Kitakami, 1938 and *D. bicarinata* Courtney, 1994 [3].

Based on relatively long antennae and postgena with microtrichia (Figure 1A and Figure 22A), the male of this species can be separated from other Asian species but similar to *D. mirabilis* following the key of Courtney (1994) [3]. However, compared to the latter, the male of *D. sinensis* sp. nov. can be separated by a glabrous oral region (Figure 22A) [3]. Further, the females of *D. sinensis* sp. nov. can be identified by the absence of digitiform setae on the distal flagellomere, this happens in all checked two females and three female heads dissected from mature pupae (Figure 4C). In contrast, *D. mirabilis* has both digitiform and sharp setae on distal antennae [3].

Pupal characters of *D. sinensis* sp. nov. are also different from the other four species. They have two pairs of spines on the mesothorax (Figure 5), in contrast to the absence of a spine in *D. nipponica* or one pair of spines in *D. sajanica* [3]. They also have no chitin bands on abdominal segments I–VII but have elongated posterior filament of gill and various dark dots on the abdomen (Figure 5 and Figure 23A–D), a combination of those characters differs from *D. tyosenensis*, *D. sajanica*, *D. bicarinata* [3,7].

Etymology: The specific epithet is an adjective referring to China, considering that it is the first new species of *Deuterophlebia* described in this country.

Distribution: CHINA (Sichuan Province).

### 3.2. Deuterophlebia yunnanensis sp. nov.

Description of male adults (Figure 9 and Figure 22C,D): Body length 2.3–2.5 mm, uniformly dark brown, most basic morphological features similar to *D. sinensis* sp. nov. (measurements here and beyond based on 2 male adults and 2 mature males dissected from pupae).

Head dark brown, flattened, width 0.43–0.48 mm. The median clypeal slightly convex, bearing about 20 setae. Compound eyes glabrous and protruded, the distance between eyes about 2.0× maximum diameter of one eye. Dense microtrichia present on the dorsal head (Figure 9B). Postgena and oral region glabrous. The ventral posterior edge of the oral region dark, sclerotized, and ridged, with an acute mental tooth in middle (Figure 9B and Figure 22B,C). 

Antenna 9.5–11.5 mm, scape broader than pedicel and about 2.0× length of the pedicel, pedicel globular, both with microtrichia on the surface. The first flagellomere subequal to the basal two segments, with a subapical tubercle and bearing 10 digitiform setae on and below the tubercle. Flagellomeres II–III similar in shape, segment II slightly longer than segment III, half in length to flagellomere I, with similar subapical tubercles and around 10 digitiform setae (Figure 9A). Flagellomere IV about 18× combined length of five basal antennal segments, and about 4× body length or 120× length of pedicel; chaetotaxy similar to *D. sinensis* sp. nov.

Thorax dark brown, covered with microtrichia; wings 5.0–6.0 mm in length, gray and semi-transparent, cubital area enlarged, anal margin and posterior half outer margin fringed with visible hair-like setae.

**Figure 9 insects-13-00593-f009:**
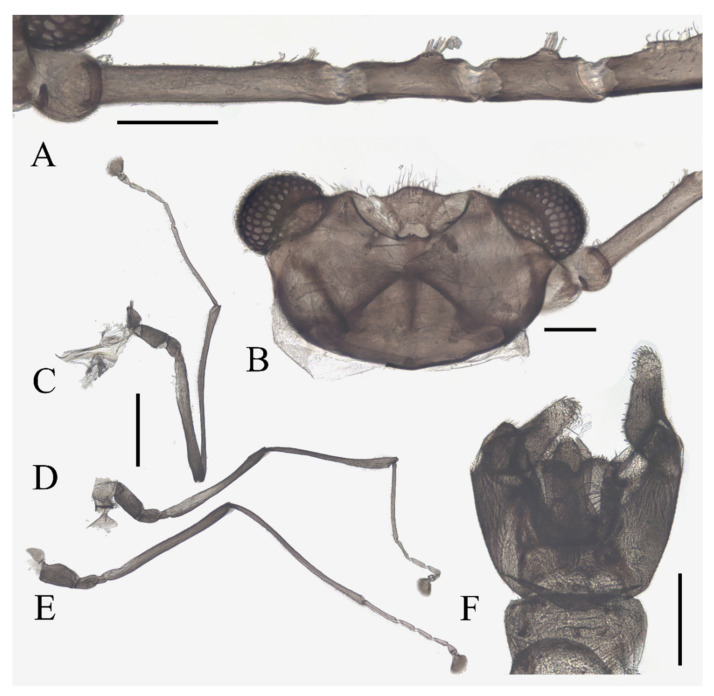
Male adult of *D. yunnanensis* sp. nov.: (**A**) male antenna; (**B**) head (ventral view); (**C**) foreleg; (**D**) midleg; (**E**) hindleg; (**F**) terminalia (dorsal view). Scale bars: (**A**,**B**,**F**) = 0.1 mm; (**C**–**E**) = 1.0 mm.

Legs showing similar chaetotaxy, with five types of setae: (1) microtrichia, densely covered on coxae, trochanters, and dorsal side of other segments; (2) bent sharp macrotrichia, sparsely on the dorsal margin of femora and proximal tibiae; (3) long and soft capitate setae, on distal half of ventral edge of tibiae, around the top of fore- and midtibiae, and ventral side of all tarsi; (4) long and straight capitate setae, only present on empodium, arranged radially; (5) short digitiform setae, 1–3 pairs on each tarsomere, mingled in the capitate setae (Figure 9C–E).

Coxae much broader than trochanters, about 2.0× latter in length. Femora:tibiae:tarsi of forelegs = 1.4:1.9:2.0; forefemora flattened, tarsomere I:II:III:IV:V = 10.0:2.5:2.0:1.8:1.2, segment V conical and glabrous; empodium shell-shaped (Figure 9C). Midleg shortest among all legs, length ratio of segments and other features similar to foreleg (Figure 9D). Femora:tibiae:tarsi of hindleg = 1.6:1.8:1.2; tarsomere I:II:III:IV:V = 4.0:1.8:1.7:1.6:1.0 (Figure 9E).

The ninth sternite almost glabrous, fused with the dorsal plate. Dorsal plate with a deep V-shaped emargination and several curved bristles. Gonostylus longer than the dorsal plate, with numerous curved sharp bristles on the flexor surface. Aedeagus subequal to gonostylus, extending beyond dorsal plate (Figure 9F).

Description of female adults (Figure 10 and Figure 11): Body length about 2.4 mm (measurements here and below based on 1 female adult and 2 mature females dissected from pupae), besides sexual differences, generally similar to the male except following features of head and legs (Figure 10).

**Figure 10 insects-13-00593-f010:**
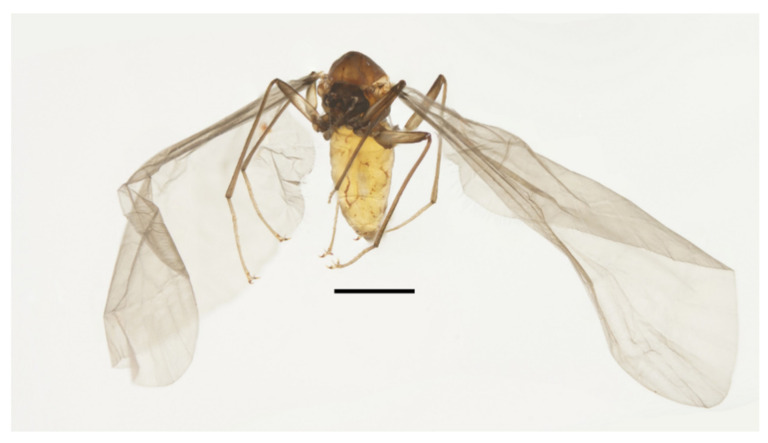
Female adult of *D. yunnanensis* sp. nov. (ventral view). Scale bar = 1.0 mm.

The oral region located near the front margin, the front margin protruded forming a smooth arc, postgena, and the oral region glabrous. Compound eyes more prominent than the male’s (Figure 11A). The antennae about 0.5 mm in length. Scape and pedicel similar to the male, antennal segment I:II:III:IV:V:VI = 2.5:1.0:2.5:1.0:1.2:1.6, flagellomeres narrowed basally and expanded apically. Flagellomeres I–III bearing 3–5 digitiform setae near the apex. Flagellomere IV with 3–5 sharp setae (Figure 11B).

**Figure 11 insects-13-00593-f011:**
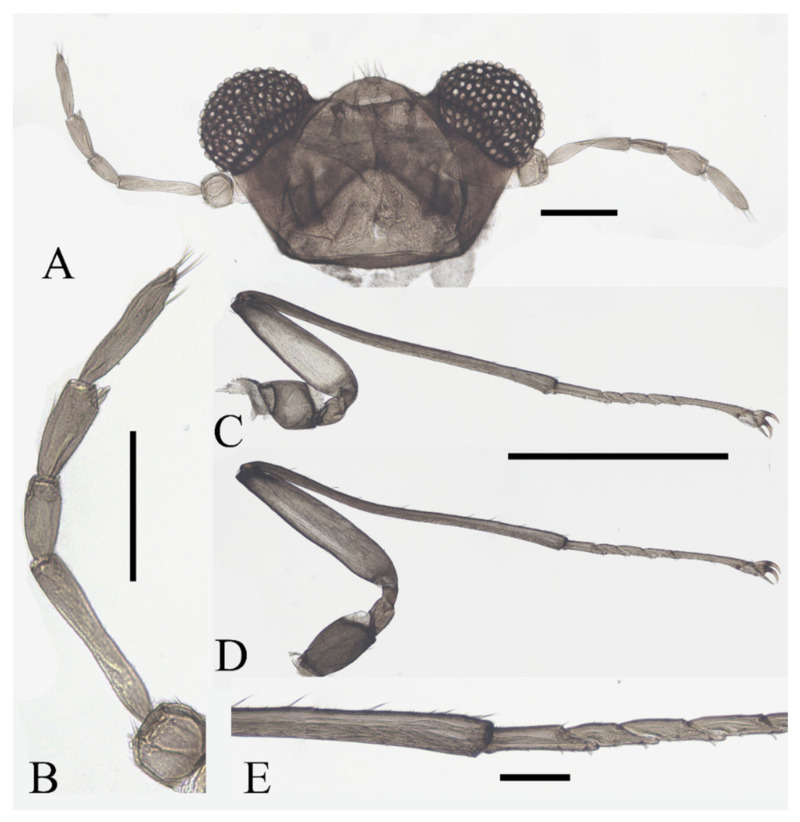
Structures of female *D. yunnanensis* sp. nov.: (**A**) head (dorsal view); (**B**) antenna; (**C**) foreleg; (**D**) hindleg; (**E**) joint of the tibia and tarsus of hindleg. Scale bars: (**A**,**B**,**E**) = 0.1 mm; (**C**,**D**) = 1.0 mm.

Legs similar; femora: tibiae: tarsi of forelegs and midlegs = 1.0:2.3:1.2, 1.3:2.3:1.2 in hindlegs. Femora wide and flat; tibiae cylindrical; length ratio of foreleg and midleg tarsomere I:II:III:IV:V = 2.2:1.0:1.1:1.1:3.2, and 1.7:1.0:1.1:1.1:3.2 in hindleg. Chaetotaxy similar, with three types of setae: (1) microtrichia, on the dorsal margin of tibiae; (2) bent sharp macrotrichia, sporadically set on femora and tibiae; (3) digitiform setae, sparsely set on the ventral side of tarsomeres I–IV, one or two pairs on each segment (Figure 11C–E).

Description of pupae (Figure 12 and Figure 23E–H): Male pupae total length 3.0–3.5 mm (measurements here and below based on 3 specimens), width about 2.1 mm, flattened, dorsal integument dark brown, eleven-segmented, most basic morphological characteristics consistent with *D. sinensis* sp. nov. (Figure 23E,F). Female pupae similar to males except for smaller body size, smaller mesothorax, and sexual differences on the ventral side, including thinner and shorter antennal sheaths and not expanded apex of leg sheaths (Figure 23G,H).

Prothorax fused with mesothorax forming the first biggest conical segment, with a median suture and two pairs of dark dots in front of paired transverse chitin bands, the length of the bands about twice wider than that of metathorax (Figure 23E,F). Both sides of the mesothorax with a spine and a gill, spines curved, originated from a round base (Figure 12A). Ventral gills dark brown, each including four filaments: posterior filament single and short, pointing backward; anterior three filaments similar in shape, slender and twisted, second and third filaments share a basal stalk, then fused with the first filament at the base (Figure 12B). Abdominal segments have numerous dark dots, concentrated on the mid-belt of each segment. The first and second abdominal segments similar, each with a pair of spine-decorated anterolateral projections, spines also exist on the lateral margin of segments VI and VII (Figure 23E,F).

**Figure 12 insects-13-00593-f012:**
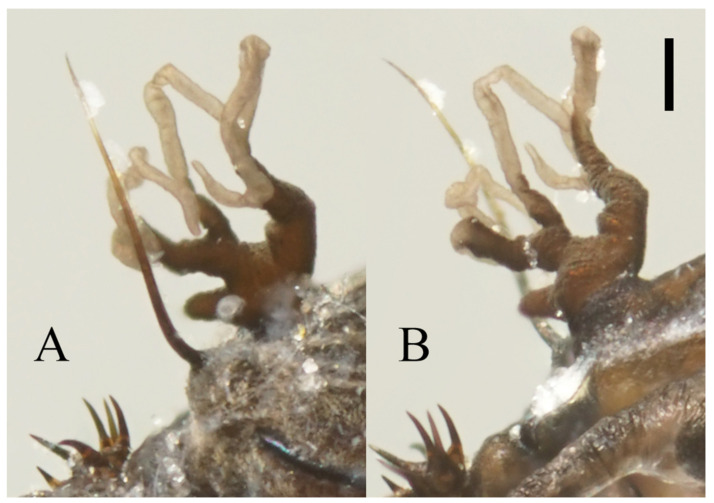
Female pupa of *D. yunnanensis* sp. nov.: (**A**) thoracic spines (dorsal view); (**B**) gill (ventral view). Scale bar = 0.1 mm.

Adult structures visible on the ventral side, male antennal sheaths extended from both sides and surrounding body 2.0 times forming an elliptic ring, male leg sheaths expanded apically, and a pair of black adhesive discs present on segments III–V respectively (Figure 23F).

Description of mature larvae (Figure 13): Most basic morphological characteristics consistent with *D. sinensis* sp. nov., body length 4.0–6.0 mm (measurements here and below based on 4 specimens), dorsal surface light brown (Figure 13A). Head almost glabrous, width about 0.60 mm, uniformly dark brown, posterior margin protruded, clypeal lobe protruded forward. Antenna two-segmented, proximal segment dark brown, longer than the ventral distal branch; distal segment bifurcated, pale, dorsal distal branch length about 5× the ventral branch (Figure 13B).

**Figure 13 insects-13-00593-f013:**
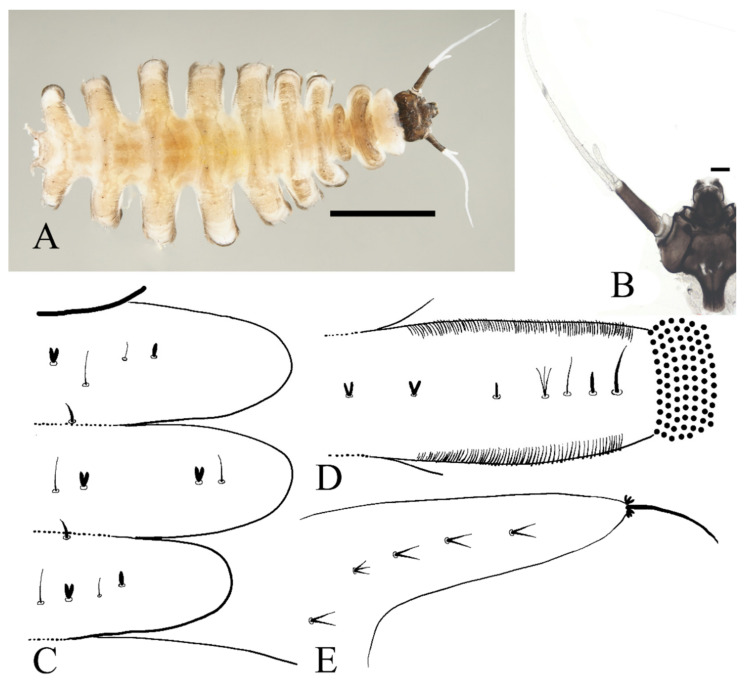
Larva of *D. yunnanensis* sp. nov.: (**A**) dorsal view; (**B**) head (dorsal view); (**C**) chaetotaxy of thorax (dorsal view); (**D**) chaetotaxy of proleg (dorsal view); (**E**) chaetotaxy of caudal appendage (dorsal view). Scale bars: (**A**) = 1.0 mm; (**B**) = 0.1 mm.

The first three segments behind the head without prolegs respectively regarded as pro-, meso- and meta-thorax. Abdomen eight-segmented, segment I–VII with a pair of pseudopodia respectively with sucker-like apex consists of 9–13 rows of tiny claws, except for the last segment which has a pair of caudal appendages instead (Figure 13A). Each margin of all pseudopodia has a row of dense hairbrush, dorsal surface of thorax and abdomen scattered with various setae (sensilla) and can be roughly divided into 4 types: (1) digitiform setae, simple or branched, some olive-shaped, on the thorax and abdominal segments; (2) long spine-like setae, only one near the top of each pseudopodium and the tip of caudal appendages; (3) hair-like setae, simple or branched, on the thorax and abdominal segments; (4) short sharp conical setae, branched, only on caudal appendages (Figure 13C–E). Chaetotaxy as in Figure 13C–E.

Material Examined: Holotype: male adult, China: Yunnan Province, Yuxi City, Ailao Mountain, 23°58′13.20″ N, 101°31′37.73″ E, 2200 m a.s.l., 7.II.2022, Xu-Hong-Yi Zheng, Long-Yi Chen, Tian-Yu Zhang leg. Paratypes: 3 males, 1 female, 4 male pupae, 4 female pupae, 4 larvae, same locality and data as holotype.

Diagnosis: Based on the key of Courtney (1994) [3], the male of this species can be keyed to three species: *D. nipponica*, *D. bicarinata*, and *D. sajanica* by relatively long antennae (longer than 8 mm) and glabrous postgena and oral region (Figure 9A,B), yet it can be distinguished from the latter two species by a distinct median clypeal lobe and less digitiform setae on flagellomeres (less than or around 10 on each segment) (Figure 9A,B) [3], from *D. nipponica* by sharp setae on dorsal hind-tibiae (Figure 9E) [7]. Female adults are separated from the three species by the length ratio of antennal segments, especially the slender terminal flagellomere and the absence of digitiform setae on it (Figure 11B) [3,7].

Pupae can be distinguished from *D. nipponica* by spines on the pupal thorax and from *D. nipponica* and *D. bicarinata* by the absence of dark bands on the pupal abdomen, which is indistinguishable from *D. sajanica* (Figure 12 and Figure 23E–H) [3].

Larvae can be separated from *D. bicarinata* and *D. sajanica* by simple inner digitiform setae on the mesothorax, and from *D. sinensis* sp. nov. by wider and bifurcated abdominal digitiform setae (Figure 8 and Figure 13C–E) [3].

Etymology: The name is an adjective referring to the type locality, which is situated in Yunnan Province of China.

Distribution: CHINA (Yunnan Province).

### 3.3. Deuterophlebia alata sp. nov.

Description of male adult (Figure 14): Only one specimen dissected from mature pupae, body length 1.8 mm, uniformly dark brown. Most basic morphological features similar to *D. sinensis* sp. nov. except for the following characters.

Head dark brown, flattened and triangular, width about 0.38 mm. Median clypeal indistinct, bearing more than 20 setae. Compound eyes glabrous, the distance between eyes about 1.5× the maximum diameter of one eye. The oral region in the form of an invaginated tubule, postgena, and oral region glabrous. Ventral posterior edge of the oral region with an acute mental tooth (Figure 14A). 

Antenna about 5.5 mm, length ratio of six articles about 3.5:1.5:3.0:2.0:2.5:110.0. Scape broader than pedicel, pedicel globular, set with microtrichia. Flagellomeres I–III bulbous distally, each with a subapical tubercle, respectively bearing around 20, 30, and 30 digitiform setae on the tubercle. Flagellomere IV about 9× combined length of five basal antennal segments or 3× body length, chaetotaxy similar to *D. sinensis* sp. nov. (Figure 14B).

Legs with similar chaetotaxy, five types of setae in total: (1) microtrichia, covered densely on all segments; (2) bent sharp macrotrichia, sparsely on dorsal tibiae; (3) long and soft capitate setae, on the ventral side of the distal tibiae and all tarsi; (4) long and straight capitate setae, only present on empodium, arranged radially; (5) short digitiform setae, 1–3 pairs on each tarsomere, mingled in the capitate setae (Figure 14D–F).

**Figure 14 insects-13-00593-f014:**
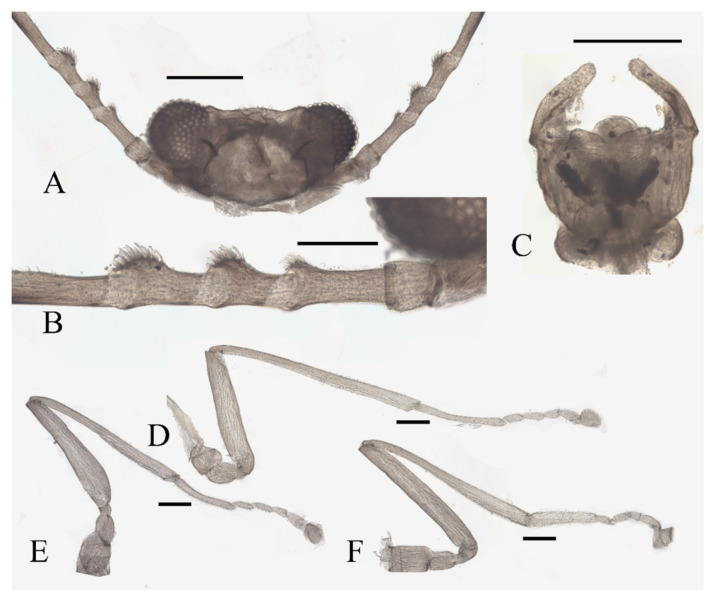
Male adult of *D. alata* sp. nov.: (**A**) head (ventral view); (**B**) male flagellomeres; (**C**) terminalia (dorsal view); (**D**) foreleg; (**E**) midleg; (**F**) hindleg. Scale bars = 0.1 mm.

Coxae broader than trochanters, equal in length in forelegs and midlegs, about 2.0× latter in hindlegs. Femora:tibiae:tarsi of forelegs = 1.0:1.6:1.3, 1.0:1.2:1.2 in hindlegs; tarsomere I:II:III:IV:V = 5.0:1.0:1.0:0.8:1.0 in forelegs and 4.0:1.0:1.0:0.8:1.0 in midlegs and hindlegs, the fourth and fifth segment conical, with microtrichia but lacking capitate setae. First tarsomere of the hindleg thickened, as wide as the apical tibia, and with a tuft of long hair-like setae at the joint (Figure 14D–F).

Abdomen nine-segmented, segment VIII reduced to a chitin ring, the ninth sternite almost glabrous, fused with the dorsal plate. Dorsal plate with a deep posteromedial notch, an acute posterolateral angle, and several curved bristles on the posterior margin. The base of the gonostylus slightly expanded outwards, the gonostylus subequal to the dorsal plate and aedeagus in length (Figure 14C).

Description of female adult (Figure 15): Only one specimen was dissected from mature pupae, body length about 1.7 mm, generally similar to the male except following features of head and legs (Figure 15).

Clypeal lobe protruded, flattened, and shovel-shaped, bearing around 10 setae at the top, narrowed at the middle forming the step-shaped lateral edge. Compound eyes more protruded than the male, wide. Postgena and oral region glabrous (Figure 15A). Antenna about 0.4 mm, article I:II:III:IV:V:VI = 2.5:1.0:1.5:1.3:1.3:1.1, flagellomeres I–III narrowed basally and expanded distally, bearing 3, 5 and 5 digitiform setae on an apical tubercle respectively, flagellomere IV with 4 digitiform setae and 3 sharp setae on apex (Figure 15B).

**Figure 15 insects-13-00593-f015:**
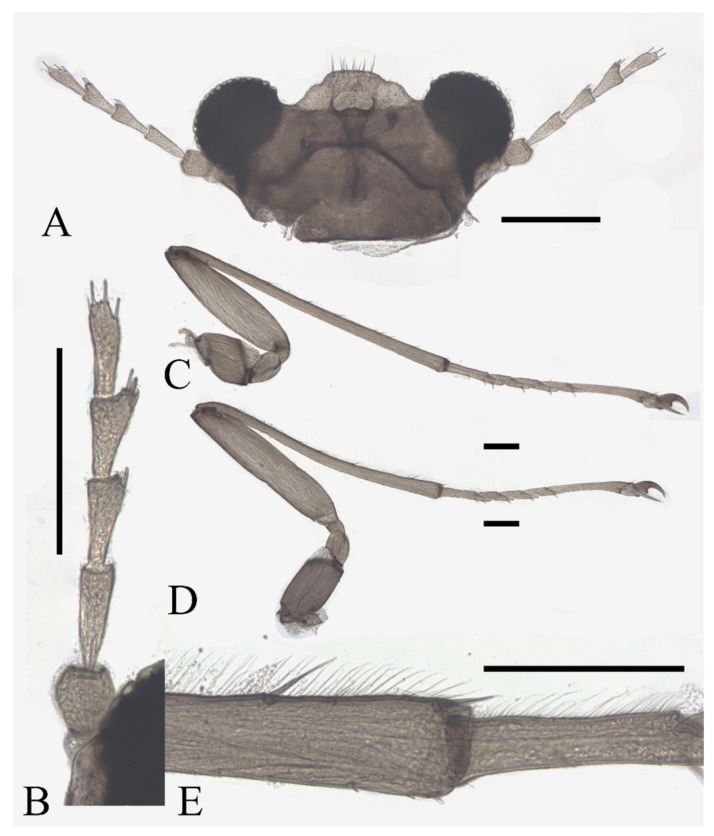
Female adult of *D. alata* sp. nov.: (**A**) head (ventral view); (**B**) female flagellomere; (**C**) foreleg; (**D**) hindleg; (**E**) joint of the tibia and tarsus of the hindleg. Scale bars = 0.1 mm.

Legs similar and have similar chaetotaxy, with three types of setae: (1) microtrichia, on the dorsal margin of tibiae and tarsomere I; (2) bent sharp macrotrichia, sporadically set on tibiae; (3) digitiform setae, sparsely set on the ventral side of tarsomeres I–IV, one or two pairs on each segment. Femora:tibiae:tarsi = 1.0:2.3:1.2 in forelegs, 1.0:1.5:1.3 in hindlegs. Length ratio of tarsomere I:II:III:IV:V of all legs = 2.0:1.0:1.0:1.0:3.0. Claws paired, sharp, and curved, with a middle sharp protrusion and a hairy spine-shaped empodium (Figure 15C–E).

Description of pupae (Figure 16 and Figure 23I–L): The male pupae length 2.1 mm and width about 1.7 mm (measurements here and below based on 2 specimens), flattened, dark brown in color, eleven-segmented, most basic morphological characteristics consistent with *D. sinensis* sp. nov. (Figure 23I,J). Female pupa similar to males except for smaller mesothorax and sexual differences on the ventral side, including thinner and shorter antennal sheaths and not expanded apex of leg sheaths (Figure 23K,L).

Prothorax fused with mesothorax forming the first biggest conical segment, with a median suture and a pair of transverse chitin bands, the bands also present on metathorax but 1/3 in length (Figure 23I,J). Both sides of mesothorax with a spine and a gill, spines twice as wide as the chitin band beside it, sharp and straight, directly originated from mesothorax without basal prominence (Figure 16A). The lateral margin of the mesothorax expanded slightly, forming a pair of thin wing-shaped outgrowths, and gills attached to the margin. Gills dark brown, each including four filaments: posterior filament single and short, pointing backward; anterior three filaments similar in shape, slender and twisted, fused at the base (Figure 16A,B).

**Figure 16 insects-13-00593-f016:**
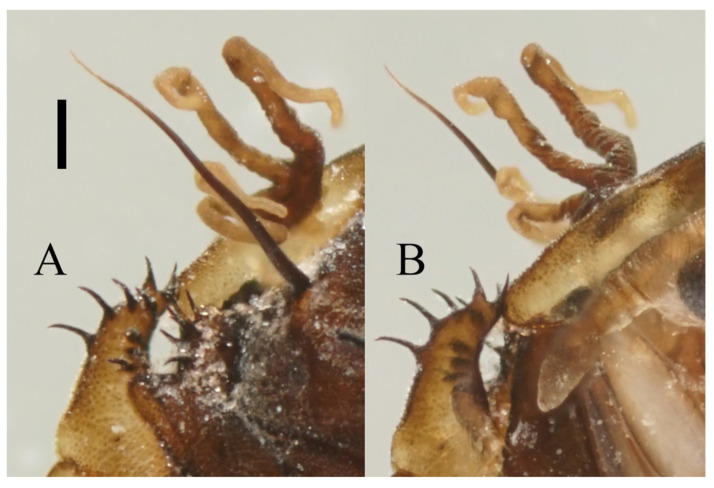
Female pupa of *D. alata* sp. nov.: (**A**) thoracic spines (dorsal view); (**B**) gill (ventral view). Scale bar = 0.1 mm.

Abdomen nine-segmented, set with numerous dark dots. The first and second abdominal segments with a pair of spine-decorated anterolateral projections, but the former pair thinner and shorter, and can be hardly seen in the ventral view. Also, spines on the lateral margin of segments VI and VII (Figure 23I,J).

Adult structures visible on the ventral side, male antennal sheaths extended from both sides, surrounding body 1.5 times forming an elliptic ring, male leg sheaths expanded apically, and a pair of black adhesive discs present on segment III–V respectively (Figure 23J).

Material Examined: Holotype: male adult (dissected from a mature pupa), China: Sichuan Province, Liangshan Prefecture, Meigu County, 28°37′6.78″ N, 101°10′52.05″ E, 2330 m a.s.l., 26.VII.2021, Peng-Xu Mu, Xu-Hong-Yi Zheng, Zhi-Teng Chen leg. Paratypes: 1 female (dissected from a mature pupa), 1 male pupae, same locality and data as holotype.

Diagnosis: The species is the smallest among the 5 species in this work. Using the key of Courtney (1994) [3], males of this species can be easily distinguished from most species by relatively short antennae (less than 7 mm) except for *D. oporina* and *D. brachyrhina*, but differ from *D. brachyrhina* by capitate setae on tibiae and tarsi (Figure 14D–F), and from *D. oporina* by the indistinct median clypeal lobe (Figure 14A). It also has the most antennal digitiform setae among known species, with a tuft of more than 20 setae on segments I–III, respectively (Figure 14B). Combining comprehensive features (shape of head and compound eyes, antennal setae, and thickened hind tibiae), it is probably closest to *D. brachyrhina* [3].

Pupal characteristics are unique, especially the lateral extension of the mesothorax which was never noticed before [1,2,3]. Other special features, including the small body size, straight and thick spines, and short bands on the metathorax, are also distinguishable.

Etymology: The specific epithet means “winged”, which refers to the wing-like extension on both sides of their pupal mesonotum.

Distribution: CHINA (Sichuan Province).

### 3.4. Deuterophlebia acutirhina sp. nov.

Description of male adults (Figure 17 and Figure 22E): Body length about 2.4 mm (measurements here and below based on 3 mature males dissected from pupae), uniformly dark brown, most basic morphological features similar to *D. sinensis* sp. nov.

Head dark brown, flattened, width 0.48 mm, deeply depressed on front margin, upper and lower ridge distinct. The median clypeal slightly convex, bearing 15–20 setae. Compound eyes glabrous, the distance between eyes about 2.0× maximum diameter of one eye, with a corner on the inner side. Microtrichia present on the dorsal head, denser and longer on the inner corner near the eye (Figure 17A). Postgena and oral region glabrous. The ventral posterior edge of the oral region sclerotized and ridged with an acute mental tooth (Figure 17A and Figure 22E). 

**Figure 17 insects-13-00593-f017:**
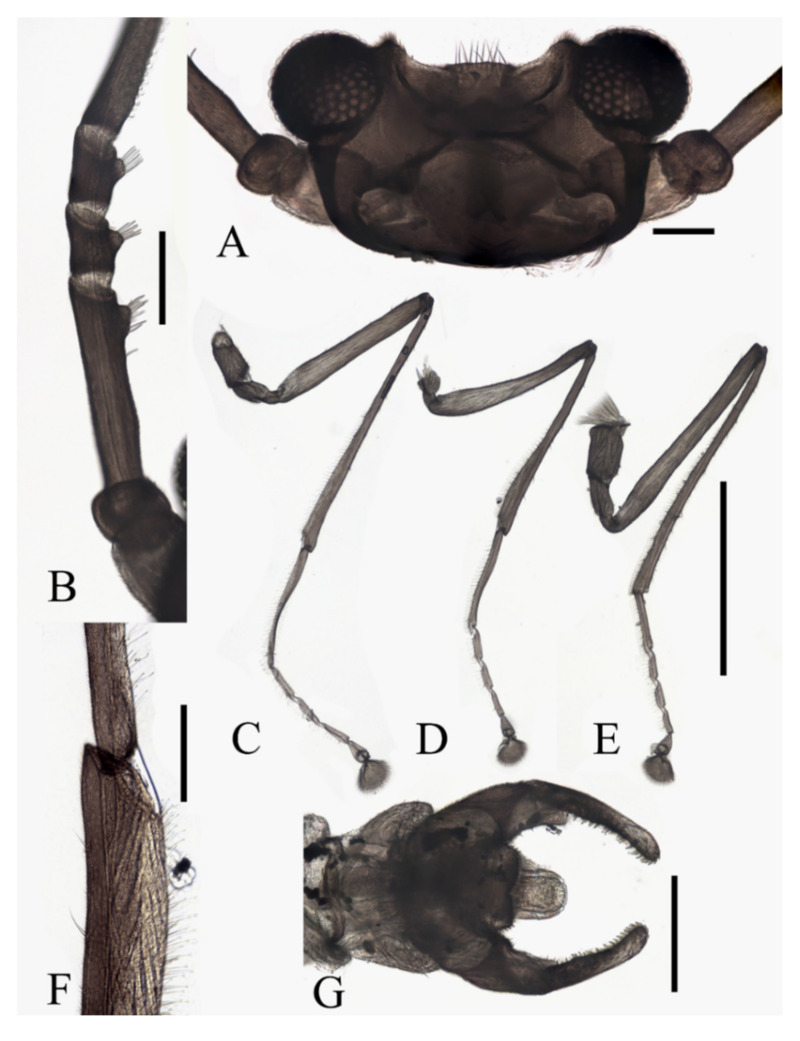
Male adult of *D. acutirhina* sp. nov.: (**A**) head (dorsal view); (**B**) male antenna; (**C**) foreleg; (**D**) midleg; (**E**) hindleg; (**F**) joint of the tibia and tarsus of midleg; (**G**) terminalia (dorsal view). Scale bars: (**A**,**B**,**F**,**G**) = 0.1 mm; (**C**–**E**) = 1.0 mm.

Antenna about 12 mm, six segments 2.0:1.0:4.0:1.0:150.0 in length. Scape broader than pedicel and about 2× length of the pedicel, pedicel globular, both with microtrichia on the surface. The first flagellomere slightly longer than the total of basal two segments, with a subapical tubercle and bearing 13 digitiform setae on and below the tubercle. Flagellomeres II–III similar in shape, with similar subapical tubercles and around 10 digitiform setae (Figure 17B). Flagellomere IV about 16× combined length of five basal antennal segments, about 4× body length or 150× length of pedicel; chaetotaxy similar to *D. sinensis* sp. nov.

The thorax dark-brown and covered with microtrichia. Legs similar, with four types of setae: (1) microtrichia, covered densely on coxae, trochanters, and dorsal tibiae; (2) bent sharp macrotrichia, sparsely on the dorsal margin of femora and basal tibiae; (3) long and soft capitate setae, on distal half of ventral tibiae and ventral side of all tarsi; (4) long and straight capitate setae, only present on empodium, arranged radially; (5) digitiform setae, 1–3 pairs on each tarsomere, mingled in the capitate setae (Figure 17C–F).

Coxae broader than trochanters, about 2.0× latter in length. Femora:tibiae:tarsi of forelegs = 1.0:1.3:1.3; forefemora flattened, tarsomere I:II:III:IV:V = 10.0:2.5:2.5:2.5:2.0, segment V conical and glabrous; empodium shell-shaped (Figure 17C). Midleg shortest, length ratio similar to foreleg (Figure 17D). Femora:tibiae:tarsi of hindleg = 1.6:1.8:1.2; tarsomere I:II:III:IV:V = 2.0:1.0:1.0:1.0:0.8 (Figure 17E).

Abdomen nine-segmented, widest at the base. Segment VIII reduced to a chitin ring. Dorsal plate with a V-shaped emargination and macrotrichia. Gonostylus longer than the dorsal plate, subequal to the aedeagus, with numerous curved bristles on the flexor surface (Figure 17G).

Description of female adult (Figure 18): Only one specimen was dissected from mature pupae, body length 2.1 mm. Oral region located near front margin, front margin protruded forming an obtuse angle with 12 setae. Postgena and oral region glabrous (Figure 18A). The antenna about 0.5 mm in length. Antennal segment I:II:III:IV:V:VI = 2.5:1.0:2.0:1.0:1.2:1.0, scape and pedicel glabrous, pedicel rounded, flagellomeres narrowed basally and expanded distally. Flagellomeres I–III each bearing 1–2 digitiform setae anteriorly near the apex. Flagellomere IV with 2 digitiform setae and 2 sharp setae (Figure 18B).

Legs similar; femora:tibiae:tarsi of forelegs and midlegs = 1.0:1.7:1.4, 1.0:1.4:1.4 in hindlegs. Femora wide and flat; tibiae cylindrical; length ratio of foreleg and midleg tarsomere I:II:III:IV:V = 2.2:1.0:1.1:1.1:3.2, and 1.7:1.0:1.1:1.1:3.2 in hindlegs. Claws paired, sharp, stout, and curved, with a sharp protrusion in the middle; empodium as a long and hairy spine. Chaetotaxy similar, with three kinds of setae: (1) microtrichia arranged on the dorsal margin of tibiae; (2) bent sharp macrotrichia, sparsely set on femora and tibiae; (3) digitiform setae, one or two pairs on each of tarsomeres I–IV (Figure 18C,D).

**Figure 18 insects-13-00593-f018:**
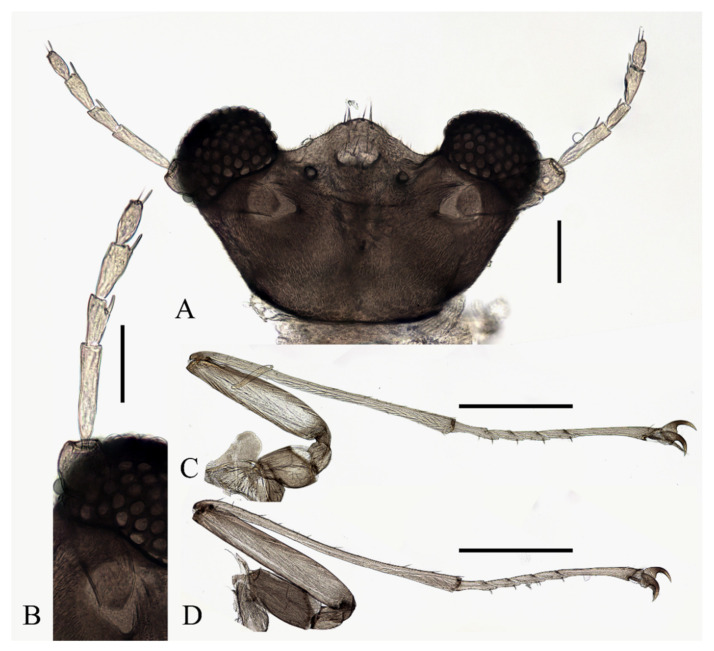
Female adult of *D. acutirhina* sp. nov.: (**A**) head (dorsal view); (**B**) female antenna; (**C**) foreleg; (**D**) hindleg. Scale bars: (**A**,**B**) = 0.1 mm; (**C**,**D**) = 1.0 mm.

Description of pupae (Figure 19 and Figure 23M–P): Male pupae length 2.9–3.3 mm, width about 2.0 mm (measurements here and below based on 4 specimens), flattened, dorsal integument dark brown, eleven-segmented, most basic morphological characteristics consistent with *D. sinensis* sp. nov. (Figure 23M,N). Female pupa similar to males, smaller in size, with smaller mesothorax and fewer dark dots. Sexual differences on the ventral side include thinner and shorter antennal sheaths and not expanded apex of leg sheaths (Figure 23O,P).

Prothorax fused with mesothorax forming the first biggest conical segment, with a median suture and two pairs of dark dots on each side, a pair of transverse chitin bands below the dots, length of the bands twice the length of those on metathorax (Figure 23M,N). Both sides of the mesothorax with a pair of spines and a gill, spines curved, originated from a round base (Figure 19A). Ventral gills dark brown, each including four filaments: posterior filament single and short and pointing backward; anterior three filaments similar in shape, slender and twisted, second and third has a common base then fused with the first filament at the base (Figure 19B). Abdominal segments with numerous dark dots, a larger pair of dots in the middle of each segment, and first and second abdominal segments with a pair of spine-decorated anterolateral projections; Spines also exist on the lateral margin of segments VI and VII (Figure 23M,N).

**Figure 19 insects-13-00593-f019:**
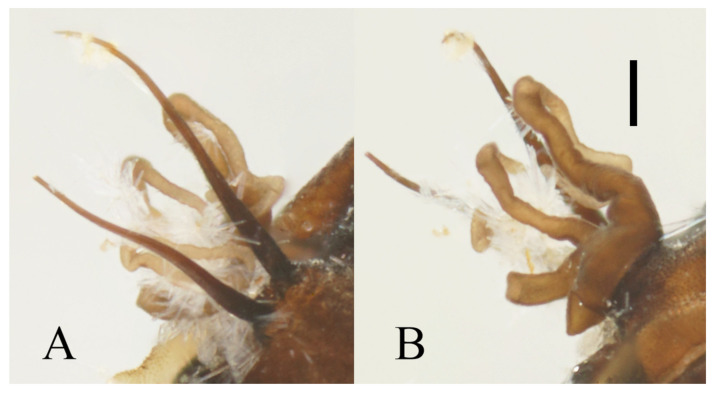
Male pupa of *D. acutirhina* sp. nov.: (**A**) thoracic spines (dorsal view); (**B**) gill (ventral view). Scale bar = 0.1 mm.

Adult structures visible on the ventral side, male antennal sheaths extended from both sides and surrounding body 2.0 times forming an elliptic ring, male leg sheaths expanded apically, and a pair of black adhesive discs present on segments III–V respectively (Figure 23N).

Material Examined: Holotype: male adult (dissected from a mature pupa), China: Fujian Province, Nanping City, Wuyishan Mountain, 27°36′4.49″ N, 117°47′21.46″ E, 331 m a.s.l., 13.III.2022, Xu-Hong-Yi Zheng leg. Paratypes: 2 males and 1 female dissected from the mature pupa, 1 male pupae, 1 female pupae, same locality and data as holotype.

Diagnosis: Based on the revision by Courtney (1994) [3], their front tubercles with more than 10 digitiform setae on flagellomere I–III and long terminal flagellomere of male adults, reminiscent of *D. sajanica*, *D. bicarinata* and *D. yunnanensis* sp. nov. in this work (Figure 17B), differ from other known species, but the absent of capitate setae around the apex of fore- and mid-tibiae can separate it from above species (Figure 17C,D,F), which was found in the Nearctic species *D. inyoensis* [2,3]. The identification of female adults mainly relies on the length ratio of antennal and leg sections and the antennal setae (Figure 18B–D). Pupae are characterized by the combination of two pairs of mesothoracic spines and large dark dots near the side of each abdominal segment (Figure 19 and Figure 23M,O), the bifurcated spines are only found in three species: *D. tyosenensis*, *D. sinensis* sp. nov. and *D. wuyiensis* sp. nov., can be separated from former two by the large abdominal dots and from the last species by the absence of abdominal dark bands [3].

Etymology: The word is a combination of the words *acuti* and *rhina*, which means “acute snout”, and refers to the acute median clypeal lobe of a female. 

Distribution: CHINA (Fujian province).

### 3.5. Deuterophlebia wuyiensis sp. nov.

Description of female adults (Figure 20 and Figure 22F): Body length 2.2–2.6 mm (Figure 20A,B) (measurements here and below based on 4 female adults). The mouth opening circular, without a mental tooth, located near the front margin. The front margin protrudes with 12–17 sharp setae, slightly depressed at the middle of the lateral edge, forming a smooth arc on the top. Postgena and oral region glabrous (Figure 20C and Figure 22F). Antenna about 0.5 mm, article I:II:III:IV:V:VI = 2.5:1.0:2.0:1.0:1.0:1.0, flagellomeres narrowed basally and rounded apically. Flagellomeres I–III each bearing 2–4 digitiform setae near the apex. Flagellomere IV with 2–4 digitiform setae and 2 sharp setae (Figure 20D).

**Figure 20 insects-13-00593-f020:**
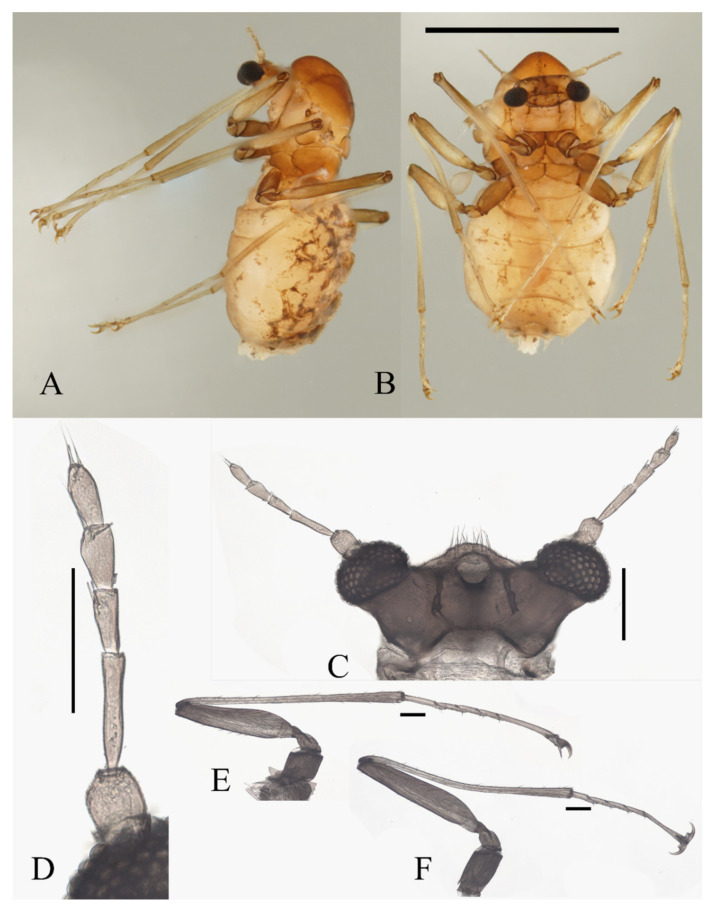
Female adult of *D. wuyiensis* sp. nov.: (**A**) body in lateral view; (**B**) body in ventral view; (**C**) head (ventral view); (**D**) antenna; (**E**) foreleg; (**F**) hindleg. Scale bars: (**A**,**B**) = 1.0 mm; (**C**–**F**) = 0.1 mm.

Legs similar; femora:tibiae:tarsi of forelegs and midlegs = 1.0:1.7:1.4, about 1.0:1.4:1.4 in hindlegs. Femora wide and flat; tibiae cylindrical; length ratio of foreleg and midleg tarsomere I:II:III:IV:V = 2.2:1.0:1.1:1.1:3.2, and 1.7:1.0:1.1:1.1:3.2 in hindlegs. Claws paired, sharp, stout, and curved, with a sharp protrusion in the middle; empodium as a long and hairy spine. Chaetotaxy of legs similar, with: (1) microtrichia arranged on the dorsal margin of tibiae; (2) bent sharp macrotrichia, sparsely set on femora and tibiae; (3) digitiform setae, one or two pairs on each of tarsomeres I–IV (Figure 20E,F).

Description of female pupae (Figure 21A,B and Figure 23Q,R): Total length 2.5–2.8 mm, width 1.7–2.0 mm (measurements here and below based on 5 specimens), flattened, brown dorsal integument, basic morphological characteristics consistent with *D. sinensis* sp. nov. (Figure 23Q,R).

Mesothorax forms the first biggest conical segment, with two pairs of dark dots on each side of the median suture, a pair of transverse chitin bands below the dots, twice the length of those on metathorax (Figure 23Q). On both sides of the mesothorax a pair of spines and a gill, spines curved. Gills brown, each including four filaments: posterior filament short, anterior three filaments slender and twisted, second and third has a common base then fused with the first filament at the base (Figure 21A). Abdominal segments scattered with dark dots, each segment has a pair of dark bands on the middle and a large pair of dots near the lateral side; first and second abdominal segments with a pair of spine-decorated anterolateral projections; spines also exist on the lateral margin of segment VI and VII (Figure 21B and Figure 23Q).

**Figure 21 insects-13-00593-f021:**
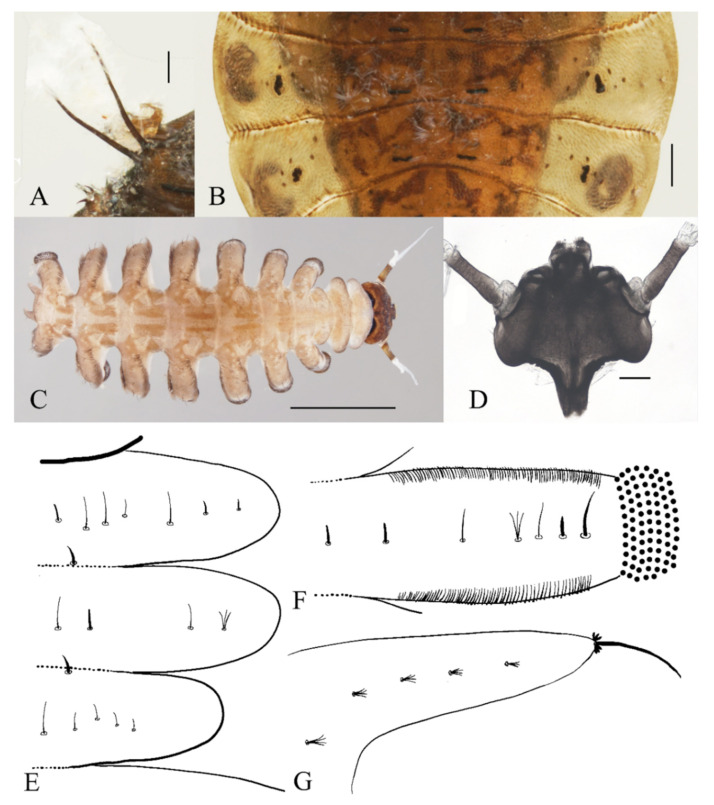
Pupal and larval characters of *D. wuyiensis* sp. nov.: (**A**) pupal thoracic spines (dorsal view); (**B**) pupal abdominal segments (dorsal view) (**C**) larva (dorsal view); (**D**) head of larva (dorsal view); (**E**) chaetotaxy of thorax (dorsal view); (**F**) chaetotaxy of proleg (dorsal view); (**G**) chaetotaxy of caudal appendage (dorsal view). Scale bars: (**A**,**B**,**D**) = 0.1 mm; (**C**) = 1.0 mm.

**Figure 22 insects-13-00593-f022:**
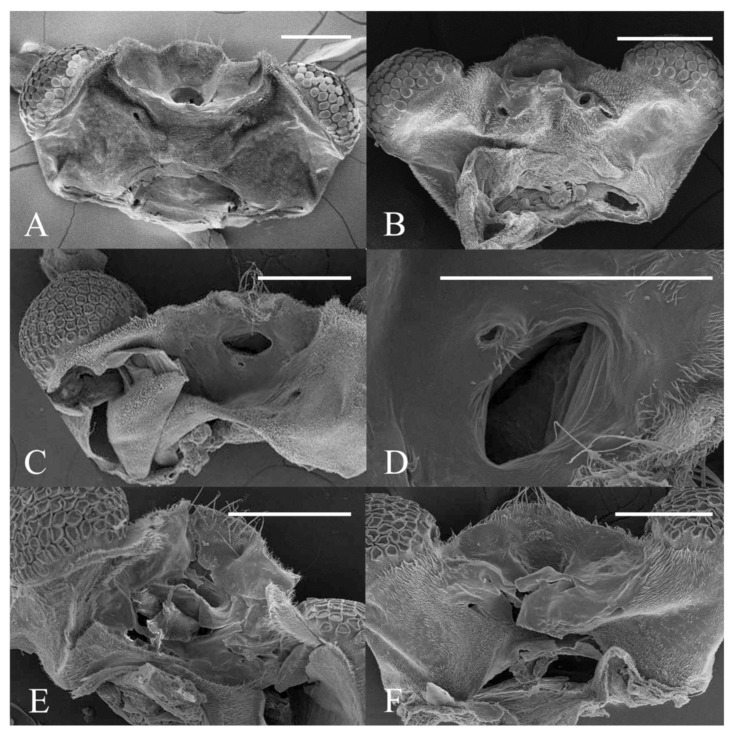
Heads of *Deuterophlebia* spp. (SEM images): (**A**) male head of *D. sinensis* sp. nov. (ventral view); (**B**) female head of *D. sinensis* sp. nov. (ventral view); (**C**) male head of *D. yunnanensis* sp. nov. (ventral view); (**D**) oral region of male *D. yunnanensis* sp. nov. (ventral view); (**E**) male head of *D. acutirhina* sp. nov. (ventral view); (**F**) female head of *D. wuyiensis* sp. nov. (ventral view). Scale bars = 0.1 mm.

Description of mature larvae (Figure 21C–G): Most basic morphological characteristics consistent with *D. sinensis* sp. nov., body length 4.0–6.0 mm (measurements here and below based on 5 specimens), dorsal surface covered with microtrichia (Figure 21C). Head almost glabrous, width about 0.60 mm, reddish-brown, posterior margin protruded, and clypeal lobe protruded, widest at posterior margin. Antenna two-segmented, basal segment dark brown, longer than the ventral distal branch; distal segment bifurcated, pale, dorsal distal branch length about 3× the ventral branch (Figure 21D).

The first three segments behind the head without prolegs respectively regarded as pro-, meso- and meta-thorax. Mesothorax widest, metathorax narrowest. Pseudopodia respectively with sucker-like apex consisting of 9–13 rows of tiny claws (Figure 21C). Each margin of all pseudopodia has a row hairbrush, dorsal surface of thorax and abdomen scattered with various setae (sensilla) and can be roughly divided into 4 types: (1) simple digitiform setae, on the thorax and abdominal segments; (2) long spine-like setae, only one near the top of each pseudopodium and the tip of caudal appendages; (3) hair-like setae, simple or trifurcated, on the thorax and abdominal segments; (4) short hair-like setae, furcated into 4–5 filaments, only on caudal appendages. Chaetotaxy as in Figure 21E–G.

Material Examined: Holotype: female adult, China: Fujian Province, Nanping City, Wuyishan Mountain, 27°44′55.52″ N, 117°40′40.77″ E, 2330 m a.s.l., 10.III.2022, Xu-Hong-Yi Zheng leg. Paratypes: 3 females, 20 female pupae, 10 larvae, same locality and data as holotype. Other material examined: 20 female pupae, 20 larvae, China: Fujian Province, Nanping City, Wuyishan Mountain, 27°52′13.33″ N, 117°51′48.69″ E, 477 m a.s.l., 14.III.2022, Xu-Hong-Yi Zheng leg.; 2 female pupae, China: Fujian Province, Nanping City, Wuyishan Mountain, 27°36′4.49″ N, 117°47′21.46″ E, 331 m a.s.l., 13.III.2022, Xu-Hong-Yi Zheng leg.

Diagnosis: Among more than 100 adults and pupae, no male was found, so the male characters of the species remained unclear. Although features of male adults were taken as the main characteristics from previous works, the special female and pupal structures of *D. acutirhina* sp. nov. are enough for their identification.

Female adults are characterized by the shape of the clypeus and the length ratio of their antennal and leg segments, which are different from all known species (Figure 21C,D) [3]. The smooth clypeus and the equal length of flagellomeres II–IV can be used to differentiate it from the sympatric species *D. acutirhina* sp. nov. (Figure 18A,B and Figure 21C,D). Pupae of the species are featured by the dark bands on all abdominal segments and two pairs of mesothoracic spines (Figure 21A,B). Similar abdominal bands only occurred in *D. nipponica* whose mesothorax are free from the spine [3,7]. Larvae can be recognized by the number and shape of dorsal setae, especially those on the prothorax, and the color and shape of their heads (Figure 21C–G).

Etymology: The name refers to its type locality, the Wuyi Mountain in East China.

Distribution: CHINA (Fujian province).

### 3.6. Molecular Study

The genetic diversity of mountain midges has never been mentioned before. We sequenced the COI gene of all the five species in this study. Due to the very limited number of specimens, only a single sample was sequenced in *D. yunnanensis* sp. nov. and *D. alata* sp. nov (Table 1).

Intraspecific genetic distances of *D. sinensis* sp. nov., *D. wuyiensis* sp. nov., and *D. acutirhina* sp. nov. are under 0.003, even between samples from sites over 300 km apart (Table 2). The distance between *D. sinensis* sp. nov. and *D. yunnanensis* sp. nov. is 0.086, while other interspecific genetic distances are between 0.148–0.175 (Table 2). The similarity between these two species is also supported by some morphological characteristics, including the long terminal flagellomere and capitate setae surrounding tibiae in males and the lack of digitiform setae on terminal flagellomere in females. We treat them as separated species, given the stable differences in all stages between them, and the existence of other related species such as *D. sajanica* and *D. bicarinata*.

### 3.7. Biological Notes

The biological habits of Chinese mountain midges are observed through the field collecting and rearing of *D. sinensis* sp. nov. and *D. wuyiensis* sp. nov. in this study. The specimens were collected from high mountains with an altitude from 2500 m to 3500 m under plateau climate in Sichuan Province of China, to the warm forests with subtropical monsoon climate in Fujian and Yunnan provinces (Figure 24). In the past hundred years, deuterophlebiid species have been discovered from mountains in the Nearctic and Palearctic regions [1,2,3,4,5,6,7,8,9,10,11,12], we provide the first Oriental report of the family by the new species *D. yunnanensis* sp. nov., *D. acutirhina* sp. nov., and *D. wuyiensis* sp. nov., this implies that they can adapt to warmer and wetter climate conditions and spread southward to the Oriental region. 

The creeks where the larvae and pupae were collected are 2–15 m wide and 0.1–1 m deep, either shaded or unshaded, containing stones of various sizes (Figure 24A). All larvae and pupae were collected from the stone surfaces; most of them were found on smooth stones of torrent areas, attached to them with their pseudopodia or discs (Figure 25A,B). No color preference for stones was found in these mountain midges.

**Figure 23 insects-13-00593-f023:**
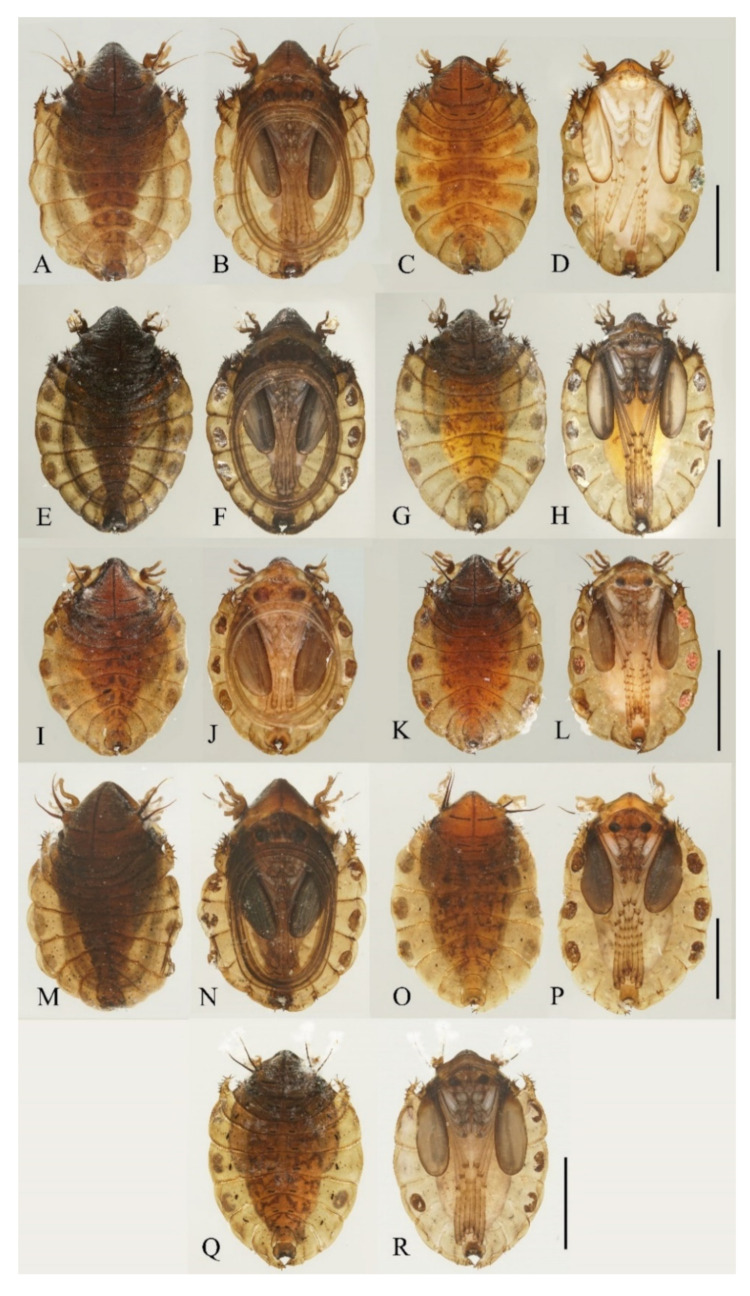
Pupae of *Deuterophlebia* spp.: (**A**) male *D. sinensis* sp. nov. (dorsal view); (**B**) male *D. sinensis* sp. nov. (ventral view); (**C**) female *D. sinensis* sp. nov. (dorsal view); (**D**) female *D. sinensis* sp. nov. (ventral view); (**E**) male *D. yunnanensis* sp. nov. (dorsal view); (**F**) male *D. yunnanensis* sp. nov. (ventral view); (**G**) female *D. yunnanensis* sp. nov. (dorsal view); (**H**) female *D. yunnanensis* sp. nov. (ventral view); (**I**) male *D. alata* sp. nov. (dorsal view); (**J**) male *D. alata* sp. nov. (ventral view); (**K**) female *D. alata* sp. nov. (dorsal view); (**L**) female *D. alata* sp. nov. (ventral view); (**M**) male *D. acutirhina* sp. nov. (dorsal view); (**N**) male *D. acutirhina* sp. nov. (ventral view); (**O**) female *D. acutirhina* sp. nov. (dorsal view); (**P**) female *D. acutirhina* sp. nov. (ventral view); (**Q**) female *D. wuyiensis* sp. nov. (dorsal view); (**R**) female *D. wuyiensis* sp. nov. (ventral view). (**C**,**D**) is recently pupated, others are mature pupae. Scale bars = 1.0 mm.

**Figure 24 insects-13-00593-f024:**
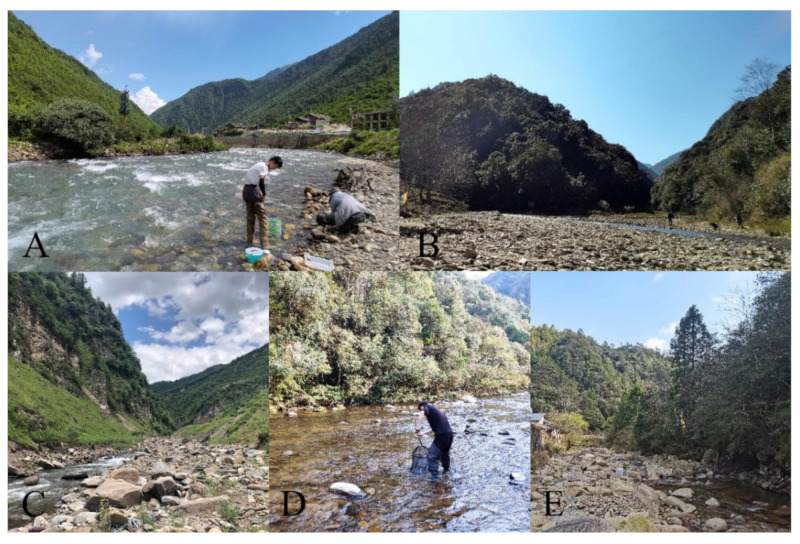
Habitat streams of *Deuterophlebia* spp.: (**A**) habitat of *D. sinensis* sp. nov. in Kangding City (Sichuan Province); (**B**) habitat of *D. yunnanensis* sp. nov. in Yuxi City (Yunnan Province); (**C**) habitat of *D. alata* sp. nov. in Meigu County (Sichuan Province); (**D**) habitat of *D. acutirhina* sp. nov. in Nanping City (Fujian Province); (**E**) habitat of *D. wuyiensis* sp. nov. in Nanping City (Fujian Province). (**C**) photo by Zhiteng Chen, (**D**) photo by Zhiming Lei, others by Xuhongyi Zheng.

During our collections of *D. sinensis* sp. nov. and *D. wuyiensis* sp. nov., both young and mature larvae, pupae, and adults can be found at the same site and time, which seems to suggest the two species as multivoltine or highly asynchronous population. To explore the emergence of *D. sinensis* sp. nov., we put some stones with pupae into rearing nets in the creek and inspected them between 7:30 and 9:30 a.m. We found two newly emerged adults during this time. Meanwhile, we searched the riverbank and found numerous dead adults on the surface of stones or in cobwebs (Figure 25D). The same phenomenon was found for the same time period for the next few days, and all discovered adults were weak or already dead. We deduce that adults emerge in the morning around the sunrise and survive only for a few hours, which is consistent with the observation by Courtney (1991) [30]. Similar phenomena are also found in *D. wuyiensis* sp. nov.

According to Courtney (1991) [30], female adults of some species may shed their wings and back to the torrent for oviposition after mating flight. Based on the discovery of wingless females in this study, the situation also occurs in *D. sinensis* sp. nov. and *D. wuyiensis* sp. nov. (Figure 25C), it may also happen in other species in this work, given the consistency of their morphology.

Our collection of *D. sinensis* sp. nov. includes nearly 100 adults, but only two female adults were caught from spider webs and one of which is wingless (Figure 6A). Seeing that the sex ratio of pupae is approaching 1:1 (more than 20 individuals checked), the imbalanced sex ratio of adults in our collection should be caused by our sampling method and their different flying and acting habits. The pupal sex ratio was also normal in *D. yunnanensis* sp. nov., *D. acutirhina* sp. nov., and *D. alata* sp. nov. However, among the three collecting sites of *D. wuyiensis* sp. nov. (coexist with *D. acutirhina* sp. nov. at one site) and over 100 pupae, no male was found. The parthenogenesis of mountain midges was first mentioned by Kitakami (1938) [7] in *D. nipponica*. He noticed that only some populations of *D. nipponica* have a normal pupal sex ratio while others only have female pupae. We found the female adults would abandon their wings right after emergence and crawl back to the water before 9:00 a.m., this further supports the parthenogenetic hypothesis in *D. wuyiensis* sp. nov.

**Figure 25 insects-13-00593-f025:**
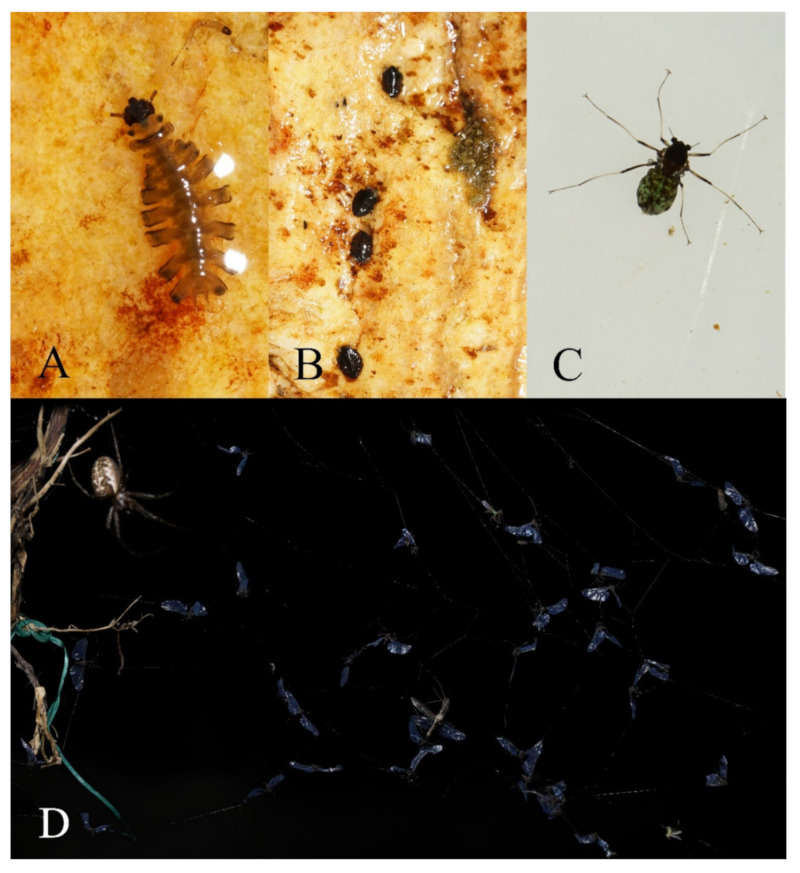
*Deuterophlebia* spp. in the nature: (**A**) larva of *D. sinensis* sp. nov.; (**B**) pupae of *D. sinensis* sp. nov.; (**C**) female adult of *D. wuyiensis* sp. nov. in water; (**D**) adults of *D. sinensis* sp. nov. on a spider web.

## 4. Discussion

Biogeographically, five new species in this study show that the family Deuterophlebiidae is distributed much more widely than previously known (Figure 26). In addition, our other collections of the genus *Deuterophlebia* (usually just one life stage, unreported) from China demonstrate that they can go far further and have more undescribed species.

**Figure 26 insects-13-00593-f026:**
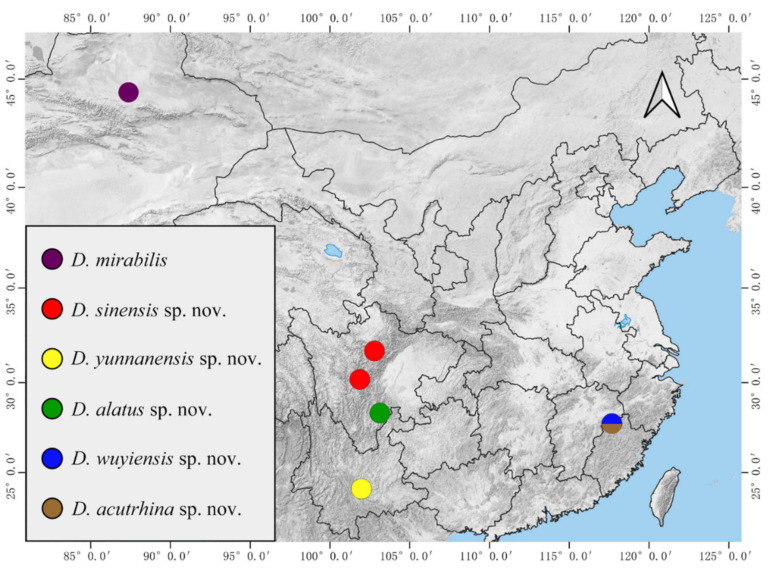
Distribution of Chinese *Deuterophlebia* spp.

At the present point, we can say with confidence that there should be more dispersal routes of the family in addition to the previous two proposed by Courtney (1994) [3]. The ancestors of extant species can go eastward first and move to Russia, Korea, and Japan directly from China instead of moving north initially. Alternatively, the species *D. inyoensis* could originate from central China and then disperse to North America because this route is much shorter and has more parsimony than the previous one from the Himalayas region.

Upon adults, pupae, and larvae described in this study, we found that female adults are more distinguishable than males in some species, and the length ratio and chaetotaxy of antennal and leg segments are more varied and usable in females than in males. Pupal and larval features are equally important, but the lacking information on most other known species hindered further research.

*D. alata* sp. nov., whose head shape, antennal length, compound eye size, and the thickening of the tarsomere I of the hindlegs in the male are similar to *D. brachyrhina* and significantly different from other species. In the phylogenetic tree reconstructed by Courtney (1994) [3] mainly using imaginal morphologies, *D. brachyrhina* was considered the basal branch of the genus with the most plesiomorphic traits. However, if we assumed that its pupal characteristics are consistent with those of *D. alata* sp. nov., its spines, and quadrifurcated gills are considered apomorphic, against the conclusion based on adults, implying problems in the old phylogenetic tree. The lack of larval and pupal specimens of Asian species and the limited morphological characteristics lead to an unstable phylogenetic tree. The reconstruction of a reliable phylogenetic tree requires more specimens of different stages and the usage of more molecular data.

The extraordinarily specialized habitat requirements of *Deuterophlebia* make themselves extremely sensitive to human activities and environment and climate-changing. In addition, updated distribution and diverse species of the family Deuterophlebiidae from China and India [5] illustrate that the taxonomy and phylogenetic reconstruction of this family needs further future research to fully decipher.

## 5. Keys to Asian *Deuterophlebia*

### 5.1. Male Adults

Modified from the key by Courtney (1994) [3]. Among the 13 Asian species, *D. tyosenensis* and *D. wuyiensis* sp. nov. are not included since male adults of them are unknown.1 Antennal length 7 mm or less, length of flagellomere IV approximately 3× the bodylength………………………………………………………………………………………….2- Antennal length 8 mm or more, length of flagellomere IV approximately 4× the bodylength (Figure 1A)……………………………………………………………………………42 Median clypeal lobe convex…………………………………………….…..…...*D. oporina*- Median clypeal lobe indistinct (Figure 14A)……………………………………………...33 Flagellomere I with more than 10 digitiform setae (Figure 14B)…...…*D. alata* sp. nov.- Flagellomere I with fewer than 10 digitiform setae………………………*D. brachyrhina*4 Mid-tibiae dorsally glabrous at the top (Figure 17F)……...……..*D. acutirhina* sp. nov.- Mid-tibiae with capitate setae around top (Figure 2E)…………………………….……55 Postgena with microtrichia (Figure 22A)…………………………………………………6- Postgena glabrous (Figure 22C)………………………………………………………...…86 Compound eyes with microtrichia between ommatidia….……………..…*D. blepharis*- Compound eyes glabrous (Figure 1B)…………………………………………………….77 Oral region with microtrichia…………………………………………….……*D. mirabilis*- Oral region glabrous (Figure 3C and Figure 22A)…………………*D. sinensis* sp. nov.8 Flagellomere I with more than 10 digitiform setae………..….*D. sajanica*, *D. bicarinata*- Flagellomere I with 10 or fewer digitiform setae…………..……………………………99 Hind-tibiae covered with sharp setae on dorsal margin……..*D. yunnanensis* sp. nov.- Hind-tibiae glabrous on dorsal margin…………………………………..…*D. nipponica*

### 5.2. Female Adults

Based on the descriptions by Courtney (1994) and other research [1,3,4,5,6,7]. Eleven species included *D. tyosenensis* and *D. brachyrhina* not included (female adults unknown).1 Flagellomere I slightly longer than Flagellomere II (Figure 15B)….…*D. alata* sp. nov.- Flagellomere I >2× longer than the length of flagellomere II (Figure 4C)……………..22 Compound eyes with microtrichia between ommatidia….………………...*D. blepharis*- Compound eyes glabrous (Figure 4D)…………………………………………………….33 Flagellomere IV without digitiform setae (Figure 4C)………………………………..…4- Flagellomere IV with several digitiform setae (Figure 18B)…………………………….54 Flagellomere IV shorter than flagellomere III (Figure 4C)…………*D. sinensis* sp. nov.- Flagellomere IV longer than flagellomere III (Figure 11B)……*D. yunnanensis* sp. nov.5 Flagellomere IV shorter than flagellomere III (Figure 18B)……………………….……6- Flagellomere IV longer or equal to flagellomere III (Figure 20D)……………………..86 Flagellomere I about 2× the length of pedicel (Figure 18B)…...…*D. acutirhina* sp. nov.- Flagellomere I about 3× the length of pedicel………………….…………………………77 Flagellomere IV shorter than the Flagellomere II…………..…………………*D. oporina*- Flagellomere IV sub-equal to Flagellomere II……………………………..…*D. mirabilis*8 Flagellomere I approximately 4× the length of pedicel…………………….*D. nipponica*- Flagellomere I shorter than 3× the length of pedicel (Figure 20D)………….…………99 Flagellomere IV with less than 6 setae (Figure 20D)…………….*D. wuyiensis* sp. nov.- Flagellomere IV with more than 3 digitiform setae……………*D. sajanica*, *D. bicarinata*

### 5.3. Pupae

Based on the descriptions by Courtney (1994) and other researchers [1,3,4,5,6,7]. Contains four known species with previously described pupal stages and five new species in this work.1 Mesothorax with lateral outgrowths (Figure 20)…….…………………*D. alata* sp. nov.- Mesothorax without lateral outgrowths (Figure 5)….…….……….…….……………...22 Mesothorax without spines on anterolateral margin….…………………... *D. nipponica*- Mesothorax with spines on anterolateral margin (Figure 5)…….…….………….……33 Mesothorax with one pair of spines on anterolateral margin (Figure 12)…….………4- Mesothorax with two pairs of spines on anterolateral margin (Figure 5)…….………54 Abdominal tergites with dark bands……….…….……….…………………*D. bicarinata*- Abdominal tergites without dark bands (Figure 23E)………………………………………………………………………………………………*D. sajanica*, *D. yunnanensis* sp. nov.5 Abdominal tergites with dark bands (Figure 21B).………………*D. wuyiensis* sp. nov.- Abdominal tergites without dark bands (Figure 23O)….…….………………….…….66 Abdominal tergites with a pair of large dark dots (Figure 23O)…*D. acutirhina* sp. nov.- Abdominal tergites without obvious larger dark dots (Figure 23A)…………….……77 Gills with elongated posterior filaments (Figure 5A)…….…….…*D. sinensis* sp. nov.- Gills with indistinct posterior filaments………………………………..…*D. tyosenensis*

Key to larvae is not given since the larvae of most species remain unclear.

## Figures and Tables

**Table 1 insects-13-00593-t001:** GenBank accession numbers of COI sequences and other information of specimens used in molecular study.

Species	Specimen Code	Life Stage	Sample Sites	GenBank Accession Number
*D. sinensis* sp. nov.	si1	Male adult	30°11′36.33″ N, 101°54′38.59″ E	ON637906
	si2	Female pupa	30°11′36.33″ N, 101°54′38.59″ E	ON637907
	si3	Male adult	31°41′21.34″ N, 102°44′35.97″ E	ON637908
*D. yunnanensis* sp. nov.	yu1	Male pupa	23°58′13.20″ N, 101°31′37.73″ E	ON637909
*D. alata* sp. nov.	al1	Male pupa	28°37′6.78″ N, 101°10′52.05″ E	ON637916
*D. acutirhina* sp. nov.	ac1	Male pupa	27°36′4.49″ N, 117°47′21.46″ E	ON637914
	ac2	Female pupa	27°36′4.49″ N, 117°47′21.46″ E	ON637915
*D. wuyiensis* sp. nov.	wu1	Female adult	27°44′55.52″ N, 117°40′40.77″ E	ON637910
	wu2	Larva	27°44′55.52″ N, 117°40′40.77″ E	ON637913
	wu3	Female pupa	27°52′13.33″ N, 117°51′48.69″ E	ON637912
	wu4	Female pupa	27°36′4.49″ N, 117°47′21.46″ E	ON637911

**Table 2 insects-13-00593-t002:** Mean values of K2P genetic distance among the DNA barcodes (COI).

Mean	*D. sinensis* sp. nov.	*D. yunnanensis* sp. nov.	*D. alata* sp. nov.	*D. acutirhina* sp. nov.
*D. yunnanensis* sp. nov.	0.086			
*D. alata* sp. nov.	0.154	0.164		
*D. acutirhina* sp. nov.	0.154	0.159	0.148	
*D. wuyiensis* sp. nov.	0.175	0.167	0.156	0.155

## Data Availability

The molecular data presented in this study are openly available in GenBank at https://www.ncbi.nlm.nih.gov/ (accessed on 23 June 2022), accession numbers ON637906–ON637916.

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
