# Peer review of "Descriptions and Barcoding of Five New Chinese Deuterophlebia Species Revealing This Genus in Both Holarctic and Oriental Realms (Diptera: Deuterophlebiidae)"

_insects, 2022, doi:10.3390/insects13070593_

Round 1

Reviewer 1 Report

The research and descriptions are professional, although I would recommend zooming in on the photos of the genitals as some of the structures are hard to see. In the future, it is better to give the main structures of the genitals in addition to photographs also in drawings.

The merit of this work is determined by the study of not only adults, but also pupae and larvae, which is very important for a correct understanding of the taxonomy and systematics of this group of aquatic insects.

Identification of new species is quite justified, but still is on the conscience of the authors.

In my opinion chapter 3.6. The Molecular Study could have been made more detailed.

Author Response

Dear reviewer:

We would like to express our sincere thanks to you and your colleagues for your careful revise and useful comments, we revised the manuscript based on your comments and edits as far as possible, but we are sorry about that we are on a collection journey in Southwest China, and some improvements require laboratory equipment cannot be completed in time. We have taken down all the useful points, such as the number of samples in molecular study and some details in photographs, all those weak sides will be solved or at least improved in future works.

Point 1: In the future, it is better to give the main structures of the genitals in addition to photographs also in drawings.

Response 1: better photographs, SEM photos or drawings will be tried in the future.

Point 2: The Molecular Study could have been made more detailed.

Response 2: this is indeed one of the biggest weak sides of this work, further molecular studies have been started, we hope they can be completed in the future and solve some problems.

Thanks again for your advices.

Sincerely yours,

Xuhongyi Zheng

Reviewer 2 Report

The MS is an important contribution and a step forward in studies of Deuterophlebiidae. The authors have collected and analysed a unique material of Deuterophlebia. It should be mentioned that the MS reviewed is the first comprehensive faunal and taxonomic review of Deuterophlebia since the revisions by Courtney (1990, 1994). The MS is mainly a taxonomic (morphological) work, but it also includes interesting ecological and geographical implications. The most important result is the morphological description and diagnoses that reveal a previously unknown, rather high species diversity of Deuterophlebiidae in China. The barcoding data serve as good additional material, which enable one to refer to these descriptions and support the validity of the described species being genetically distinct from each other.

On the other hand, the MS has several evident gaps and weak sides. Some points of criticism are listed below.

The publications on the morphology, bionomics and distribution of Palaearctic species of Deuterophlebia and used and cited incompletely, especially those on D. mirabilis and D. sajanica.

The photos of morphological details, although informative, are unfortunately not of very high quality. It seems that many photos were taken from the adult specimens that were not enough cleared in KOH, so the male hypopygia, heads, antennae etc. are not transparent (not cleared from tissues) and many details are less visible than they should be. Hence, it would be essential to provide better photos, as far as possible.  Possibly it would be necessary to make better permanent slides of adults and to use contrasting techniques like phase contrast or DIC.

The COI sequences were obtained only for a few specimens (one to four for each of five species, and eleven specimens in total). This number is surprisingly low as compared to the total number of specimens examined.   

It would be interesting and important to provide more information on the collecting habitats, i.e. to provide photos and to characterize the conditions of streams (temperature, current velocity, water regime, trophic status etc.).

The text is prepared and written not very carefully, with some misprints and gaps (some of which were corrected in the PDF by the reviewer). Spelling of two species names proposed is to be refined (alata should be instead of alatus, acutirhina instead of acutrhina), as well as the etymology (see the commented MS).

Further suggestions and suggested minor corrections to the text are available in the commented PDF of the MS.

Author Response

Dear reviewer:

We would like to express our sincere thanks to you and your colleagues for your careful revise and useful comments, we revised the manuscript based on your comments and edits as far as possible, but we are sorry about that we are on a collection journey in Southwest China, and some improvements require laboratory equipment cannot be completed in time. We have taken down all the useful points, such as the number of samples in molecular study and some details in photographs, all those weak sides will be solved or at least improved in future works.

Point 1: The publications on the morphology, bionomics and distribution of Palaearctic species of Deuterophlebia and used and cited incompletely, especially those on D. mirabilis and D. sajanica.

Response 1: revised. We feel sorry for overlooking their distribution Kyrgyzstan, it is obvious in the map given by Courtney 1994.

Point 2: The photos of morphological details, although informative, are unfortunately not of very high quality. It seems that many photos were taken from the adult specimens that were not enough cleared in KOH, so the male hypopygia, heads, antennae etc. are not transparent (not cleared from tissues) and many details are less visible than they should be. Hence, it would be essential to provide better photos, as far as possible. Possibly it would be necessary to make better permanent slides of adults and to use contrasting techniques like phase contrast or DIC.

Response 2: we have tried our best to obtain good pictures using our existing equipment, but some of them are still imperfect, due to a combination reason of the lacking of experience in handling tiny specimens, our equipment and a limited number of specimens (especially in D. alata sp. nov.). We also tried some permanent slides, but with too much bubbles. We are keep on trying to get better photographs, SEM photos or drawings will also be tried in the future.

Point 3: The COI sequences were obtained only for a few specimens (one to four for each of five species, and eleven specimens in total). This number is surprisingly low as compared to the total number of specimens examined.

Response 3: this is indeed one of the biggest weak sides of this work, further molecular studies have been started, we hope they can be completed in the future and solve some problems.

Point 4: It would be interesting and important to provide more information on the collecting habitats, i.e. to provide photos and to characterize the conditions of streams (temperature, current velocity, water regime, trophic status etc.).

Response 4: figures of different habitat streams are given in the revised version. We didn’t payed much attention to the conditions of streams, which is a great pity. They will be mentioned in future works.

Point 5: The text is prepared and written not very carefully, with some misprints and gaps (some of which were corrected in the PDF by the reviewer). Spelling of two species names proposed is to be refined (alata should be instead of alatus, acutirhina instead of acutrhina), as well as the etymology (see the commented MS).

Response 5: We apologize for making those stupid mistakes. All of them have been corrected now.

Other suggestions and minor corrections are all corrected in the text. In addition, we delete the sentences about the male terminalia of D. mirabilis in the diagnosis and key to D. sinensis sp. nov., because we found some conflicts between the original description and Courtney’s description of this species. Thanks again for your advices.

Sincerely yours,

Xuhongyi Zheng

Reviewer 3 Report

Authors present a very nice and important study of the dipteran family Deuterophlebiidae. This manuscript uses an integrated approach based on COI sequences, morphology, and ecology to examine this group. Information gathering from the present manuscript will be very useful for the further taxonomic study of this family. 

Author Response

Dear reviewer:

We would like to express our sincere thanks to you and your colleagues for your careful revise and useful comments, we revised the manuscript based on your comments and edits as far as possible, but we are sorry about that we are on a collection journey in Southwest China, and some improvements require laboratory equipment cannot be completed in time. We have taken down all the useful points, such as the number of samples in molecular study and some details in photographs, all those weak sides will be solved or at least improved in future works.

Here are responses for the points you mentioned in the text, we also explained them in the text. Total genomic DNA was extracted from the abdomen of specimens, most important characters on head, legs and terminalia are not destructed. Preservation of specimens were mentioned in Materials and Methods, “one male adult paratype of D. sinensis sp. nov., D. yunnanensis sp. nov., D. acutirhina sp. nov. and one female adult paratype of D. wuyiensis sp. nov. mounted into slides by neutral balsam, deposited in the Diptera collection of College of Life Sciences, Nanjing Normal University, other specimens preserved in 85% ethanol and deposited in the Diptera collection of College of Life Sciences, Nanjing Normal University and School of Grain Science and Technology, Jiangsu University of Science and Technology”.

We have considered your important suggestions for diagnosis, they are good and constructive, but we find that sometimes it is difficult to separate “diagnostic characters” and “taxonomic notes”. Our version separated sections by life stages, compare related species and analyze their characters at the same time, might be more logical and clearly.

In addition, we delete the sentences about the male terminalia of D. mirabilis in the diagnosis and key to D. sinensis sp. nov., because we found some conflicts between the original description and Courtney’s description of this species.

Other suggestions and minor corrections are all corrected in the text. Thanks again for your advices.

Sincerely yours,

Xuhongyi Zheng